# DISENTANGLED GANS FOR CONTROLLABLE GENERATION OF HIGH-RESOLUTION IMAGES

## ABSTRACT

Generative adversarial networks (GANs) have achieved great success at generating realistic samples. However, achieving disentangled and controllable generation still remains challenging for GANs, especially in the high-resolution image domain. Motivated by this, we introduce *AC-StyleGAN*, a combination of AC-GAN and StyleGAN, for demonstrating that very limited supervision is sufficient to benefit the *generic* disentanglement learning of high-resolution images. Specifically, by only using 1% of the labelled data, the disentanglement quality is very close to the fully supervised case and significantly outperforms the unsupervised alternative. Inspired by the observed separation of fine and coarse styles in Style-GAN, we then extend AC-StyleGAN to a new image-to-image model called *FC-StyleGAN* for semantic manipulation of fine-grained factors in a high-resolution image. In experiments, we show that FC-StyleGAN performs well in only controlling fine-grained factors, with the use of instance normalization, and also demonstrate its good generalization ability to unseen images. Finally, we create two new datasets – *Falcor3D* and *Isaac3D* with higher resolution, more photorealism, and richer variation, as compared to existing disentanglement datasets.

## 1 INTRODUCTION

High-resolution controllable generation is an important component in many applications, such as image editing (Yao et al., 2018), 3D scene understanding (Eslami et al., 2018) and inverse graphics (Kulkarni et al., 2015). Generative adversarial networks (GANs) (Goodfellow et al., 2014) have achieved great success at generating realistic images, of which two representatives are Style-GAN (Karras et al., 2019) for unconditional generation and BigGAN (Brock et al., 2018) for class conditional generation. However, the controllable generation is still a challenge for state-of-the-art GANs, especially with high-resolution images. For instance, StyleGAN cannot be directly used to synthesize high-fidelity human faces by specifically controlling skin color or eye size without affecting other face attributes. Also, BigGAN is not able to control the hair color or length of a dog image without changing other features.

Disentanglement of various factors allows us to independently control the variations across all the factors. But this is not easy to learn in GANs without further modifications such as adding regularization to encourage better disentanglement. For example, InfoGAN (Chen et al., 2016) proposes maximizing mutual information between latent code and its reconstruction for an unsupervised disentanglement. Most recently, Locatello et al. (2019a) has showed that unsupervised disentanglement learning is impossible without *model inductive bias* or *supervision*. In that light, several recent works attempt to learn disentanglement in GANs by introducing a strong model bias (Nguyen-Phuoc et al., 2019). However, these works are restricted to a specific domain (e.g. 3D rotations only) and usually difficult to scale up to higher resolution, mainly due to the intrinsic limitations of the model biases themselves. This motivates us to study the impact of limited supervision in a *generic* disentanglement learning with GANs, especially in the more challenging high-resolution image domain.

**Main contributions:** By combining the advantages of AC-GAN (Odena et al., 2017) and Style-GAN, we introduce AC-StyleGAN, and demonstrate that, very limited supervision is sufficient: just 1% of the labelled data significantly improves the disentanglement quality, as compared to the unsupervised alternatives. However, if we are only interested in controlling a *subset* of factors (effectively treating the rest as random unobserved nuisance variables), we find that the disentanglement qual-

ity of AC-StyleGAN degrades significantly. We believe this is due to the latent nuisance factors strongly confounding the observed factors, a common and difficult problem in high-dimensional partially observed latent variables models (Bishop, 1998).

To address this, we propose FC-StyleGAN, a new image-to-image model that adds controllable fine-grained factors along with a super-resolution process. This is inspired by the separation of fine and coarse styles, observed in StyleGAN. FC-StyleGAN enables semantic manipulation of fine styles in high-resolution images, without a commonly-used encoder-decoder structure.

Finally, we create two new high-quality datasets – *Falcor3D* and *Isaac3D* – that present a new challenge for controllable generation in terms of image resolution, photorealism, and richness of style factors, as compared to existing disentanglement datasets such as 3D Chairs (Aubry et al., 2014), dSprite (Matthey et al., 2017) and MPI3D (Gondal et al., 2019).

Thus, we propose new semi-supervised GAN architectures that enable a *generic* disentanglement learning of high-resolution images as well as new high-quality disentanglement datasets.

## 2 RELATED WORK

**Disentanglement learning.** Learning *generic* disentangled representations in an unsupervised way has attracted a lot of attention. Two representative models are InfoGAN (Chen et al., 2016) and $\beta$-VAE (Higgins et al., 2017). The nature of unsupervised disentanglement learning does not guarantee that the learned disentangled factors are semantically meaningful without an additional inductive bias or supervision (Locatello et al., 2019a; Nguyen-Phuoc et al., 2019; Locatello et al., 2019b). Another line of work aims to learn disentangled representations via supervised learning wherein factors are observed variables (Kulkarni et al., 2015; Reed et al., 2014; Xiao et al., 2018; Locatello et al., 2019b). However, little work has systematically investigated the necessity of *very limited supervision* to *generic* disentanglement learning. The analysis of Locatello et al. (2019b) only applies to disentangled VAEs on low-resolution datasets. Instead, our work tries to answer a more practical question: with very little supervision on more complex and higher-resolution datasets, can we still well disentangle the factors of variation while maintaining the high generation quality?

**Deep image manipulation.** Deep neural networks have enabled various image editing tasks, such as style transfer (Gatys et al., 2016), image-to-image translation (Zhu et al., 2017), automatic colorization (Zhang et al., 2016) and 3D-aware attribute editing (Yao et al., 2018). None of the above methods except the 3D-SDN (Yao et al., 2018) has dealt with the semantic manipulation of multiple attributes for scene images. Different from the 3D-SDN and other previous works on attribute editing that have been mostly focused on the 3D geometry manipulation, our proposed FC-StyleGAN is designed to semantically manipulate the fine-grained factors instead, such as lighting conditions and object colors. Furthermore, FC-StyleGAN does not apply an encoder-decoder structure as commonly used in previous works for semantic manipulation (Yao et al., 2018), instead, it adds controllable fine-grained factors along with a super-resolution process, which turns out to perform well in editing fine styles of high-resolution images with a good generalization ability.

## 3 MODELS

### 3.1 BACKGROUND ON AC-GAN AND STYLEGAN

**AC-GAN.** AC-GAN (Odena et al., 2017) is a variant of class conditional GANs. In the AC-GAN, the generator inputs are a latent $z$ (or called random noise) and the class label. The input to the discriminator is the real or fake image, whilst its output is the probability that the image is real and the prediction of the class label. Odena et al. (2017) has shown that this modification to the standard conditional GAN formulation produces more realistic images and appears to stabilize training.

**StyleGAN** StyleGAN (Karras et al., 2019) is a state-of-the-art GAN architecture for unsupervised image generation, particularly for high-fidelity human faces. Basically, StyleGAN comprises a mapping network whose role is to map the latent $z$ to an intermediate space, which then controls the styles at each convolutional layer in the synthesis network via adaptive instance normalization (AdaIN) (Ulyanov et al., 2016; Huang & Belongie, 2017). StyleGAN also enables the separation of fine-grained and coarse-grained features. For example, modifying the styles of low-resolution blocks affects only coarse-grained features (e.g. pose and eyeglasses), while modifying the styles of high-resolution blocks affects only fine-grained features (e.g. color scheme and microstructure).

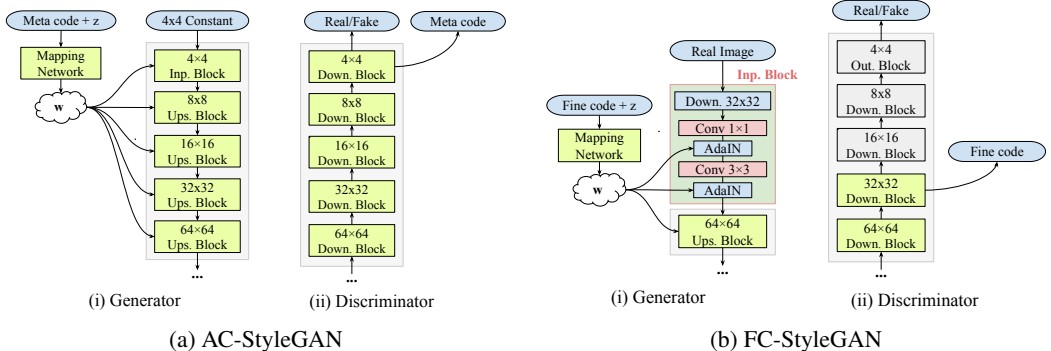

Figure 1: An overview of model architectures (a) AC-StyleGAN and (b) FC-StyleGAN. (a) The generator conditions on meta code for generation and the discriminator predicts its value. (b) We downsample the real image into 32x32 resolution and replace the lower resolution blocks (4x4 - 32x32) in the AC-StyleGAN generator by a new input block. Also, the discriminator predicts the value of fine code from the 32x32 block instead.

## 3.2 AC-STYLEGAN FOR CONTROLLABLE IMAGE SYNTHESIS

We introduce AC-StyleGAN, a combination of AC-GAN and StyleGAN, that enables conditional generation of high-resolution images. As shown in Figure 1a, the generator in AC-StyleGAN conditions on a meta code by simply concatenating the meta code, a vector representing all the factors of variation, with the latent $z$. The discriminator in AC-StyleGAN now provides two outputs, the classification of real/fake and the prediction of meta code. And finally, the outputs of the mapping network – the conditioned styles – will modulate each block in the synthesis network via AdaIN. We choose AC-GAN, instead of other conditional GAN methods, such as cGANs (with Projection Discriminator) (Miyato & Koyama, 2018), mainly because only AC-GAN has the property of reconstructing meta code in the discriminator, which can easily be extended to a semi-supervised disentanglement learning framework.

More formally, let $x_r^{(n)}$ denote the $n$-th image from the set of $N$ real images and let $c_r^{(n)}$ be the corresponding meta code, randomly sampled from the dataset. Let $c_f^{(n)}$ denote a random meta code sampled from the label distribution, and let $D(\cdot)$ and $G(\cdot)$ denote the discriminator and generator neural networks. We also assume $s_r^{(n)}$ and $s_f^{(n)}$ represent the real and fake classification logits in the discriminator, respectively, and $\hat{c}_r^{(n)}$ and $\hat{c}_f^{(n)}$ denote the predictions of $c_r^{(n)}$ and $c_f^{(n)}$, respectively. From Figure 1a, we have

$$(s_r^{(n)}, \hat{c}_r^{(n)}) = D(x_r^{(n)}) \quad \text{and} \quad (s_f^{(n)}, \hat{c}_f^{(n)}) = D(G(c_f^{(n)}, z^{(n)})) \tag{1}$$

Since we are aiming at achieving both high image quality and good controllability, we define a semi-supervised loss function for the generator and discriminator in AC-StyleGAN as follows,

$$L^{(G)}(G, D) = \frac{1}{N} \sum_{n=0}^{N-1} \underbrace{l_{\text{GAN}}(s_r^{(n)}, s_f^{(n)})}_{\text{GAN loss}} + \gamma \underbrace{\|\hat{c}_f^{(n)} - c_f^{(n)}\|}_{\text{unsupervised disentanglement}}$$

$$L^{(D)}(G, D) = -\frac{1}{N} \sum_{n=0}^{N-1} \underbrace{l_{\text{GAN}}(s_r^{(n)}, s_f^{(n)})}_{\text{GAN loss}} + \gamma \underbrace{\|\hat{c}_f^{(n)} - c_f^{(n)}\|}_{\text{unsupervised disentanglement}} + \frac{N\eta^{(n)}\gamma}{\sum_{n=0}^{N-1} \eta^{(n)}} \underbrace{\|\hat{c}_r^{(n)} - c_r^{(n)}\|}_{\text{supervised disentanglement}}$$

$$\tag{2}$$

which consists of three terms: the GAN loss term, unsupervised disentanglement term and supervised disentanglement term. For the GAN loss, we apply the non-saturating loss plus gradient penalty in the discriminator, as in StyleGAN. For both the unsupervised and supervised disentanglement terms, we simply use $l_2$ norm of the reconstruction losses.

Furthermore, we introduce two coefficients $\gamma$ and $\eta^{(n)}$ in the above loss function: (i) the disentanglement coefficient $\gamma \in (0, \infty)$ balances the trade-off between the generation quality and disentanglement performance; (ii) the coefficient $\eta^{(n)}$ is the *label mask* denoted by $\eta^{(n)} = \mathbf{1}_{\{\alpha^{(n)} < \alpha\}}$

where $\alpha^{(n)} \overset{i.i.d.}{\sim}$ Uniform$[0, 1]$ and $\alpha \in [0, 1]$ is a hyperparameter. Accordingly, the hyperparameter $\alpha$ controls the fraction of labelled data that will be used for supervision (*fully unsupervised* in the disentanglement learning if $\alpha = 0$, and *fully supervised* if $\alpha = 1$).

Note that we sample all the $\alpha^{(n)}$'s before training so that the label mask $\eta^{(n)}$ remains unchanged during training to reflect the real semi-supervised case. We also apply a rescaling operation by multiplying the supervised disentanglement term in Eq. (2) with the rescaling coefficient $\frac{N}{\sum_{n=0}^{N-1} \eta^{(n)}}$ to take an average of effective supervised signals within a minibatch. Finally, note that AC-StyleGAN reduces to an InfoGAN variant in the special case of $\alpha = 0$ (fully unsupervised), when an appropriate reconstruction loss is used.

### 3.3 FC-StyleGAN for Semantic Manipulation of Fine Styles

While we apply AC-StyleGAN with supervision over all observed factors for demonstrating a controllable generation of high-resolution images in GANs, in a potentially more realistic case where we are only interested in controlling a *subset* of factors (effectively treating the rest as random unobserved nuisance variables), the disentanglement quality of AC-StyelGAN drops significantly, as shown in Section 4.2. Furthermore, AC-StyleGAN cannot be directly applied to manipulate an existing high-resolution image. Therefore, we propose a new image-to-image model called FC-StyleAGN (i.e., Fine-grained Controlled StyleGAN) for only controlling fine-grained factors.

Inspired by the observations that the lower resolution blocks in the StyleGAN generator learn the coarse-grained features while its high-resolution blocks accounts for fine styles, the generator in FC-StyleGAN does not contain the lower-resolution blocks. As shown in Figure 1b, the generator instead takes the real image as one of its inputs by downscaling it to a lower resolution $\phi$ (e.g., $\phi = 32$ in Figure 1b). After that, it generates the high-resolution image by only modulating the fine-grained code into higher resolution blocks. Also, the discriminator in FC-StyleGAN predicts the value of fine-grained code from the block with resolution $\phi$, instead of the last output block. The intuition is that similar to the observations in the generator, lower resolution blocks in the discriminator have less relationship with fine-grained styles. So it makes more sense to directly predict the meta code from higher resolution blocks. Note that the value of $\phi$ varies from 4 to image size, determining what factors are identified as fine-grained. Finally, the loss function in FC-StyleGAN is the same with Eq. (2) in AC-StyleGAN, indicating a semi-supervised approach.

During training of FC-StyleGAN, the generator takes the downscaled image as is, and just *embellishes* it by increasing the resolution and generating the missing pixel-level detail. As it does not need to learn the coarse-grained features any more, presumably it will have much easier time handling complex high-resolution images. Furthermore, since the generator is still style-based in higher-resolution blocks with fine-grained code as its input, we will retain the control over fine styles. However, there are two caveats in FC-StyleGAN. The first caveat is that as an image-to-image model, the original fine-grained factors in the input image may interference the control of fine code over the fine styles in the output image. In such sense, we emphasize the role of inductive biases on the model – the use of instance normalization in the generator. Because the output of instance normalization preserves the spatial structure of image content by only normalizing fine styles (Ulyanov et al., 2016), we argue that the original fine styles in the input image could be washed away by instance normalization, and thus the fine code will fully control the fine-grained factors.

The second caveat is how to identify fine-grained factors in a dataset before training FC-StyleGAN. To this end, we use the meta code of all factors as the input of FC-StyleGAN with different downscaled resolutions $\phi$. For each $\phi$, we introduce a new term – *interpolation variance* of each factor $c_i$, denoted by $\beta_i(\phi)$, based on which the factor $c_i$ is defined as *fine-grained at $\phi$* if its interpolation variance satisfies that $\beta_i(\phi) > \beta_0$, where $\beta_0$ is a pre-defined threshold. The interpolation variance $\beta_i(\phi)$ is calculated as follows: Given $N$ real images, we do the latent traversal over each factor $c_i^{(n)}$ of an image $I^{(n)}$ to get $S$ interpolated (fake) images $\{\hat{I}_{i,0}^{(n)}, \cdots, \hat{I}_{i,S-1}^{(N)}\}$. Each interpolated image $\hat{I}_{i,s}^{(n)}$ is then fed into the discriminator to get a predicted factor $\hat{c}_{i,s}^{(n)}$. Thus, the interpolation variance is the average variance of $\hat{c}_{i,s}^{(n)}$ over the interpolation $s$-dimension, which is

$$\beta_i(\phi) = \frac{1}{N} \sum_{n=0}^{N-1} \text{Var}_s \left[ \hat{c}_{i,s}^{(n)} \right] \tag{3}$$

| Datasets | # of Images | # of Factors | Resolution | 3D |
|----------|-------------|--------------|------------|-----|
| dSprites | 737,280 | 5 | 64x64 | ✗ |
| Noisy dSprites | 737,280 | 7 | 64x64 | ✗ |
| Scream dSprites | 737,280 | 7 | 64x64 | ✗ |
| SmallNORB | 48,600 | 5 | 128x128 | ✓ |
| Cars3D | 17,568 | 3 | 64x64 | ✓ |
| 3dshapes | 480,000 | 7 | 64x64 | ✓ |
| MPI3D | 640,800 | 7 | 64x64 | ✓ |
| *Falcor3D* | 233,280 | 7 | 1024x1024 | ✓ |
| *Isaac3D* | 737,280 | 9 | 512x512 | ✓ |

Table 1: Summary of the proposed two datasets, compared with currently commonly-used datasets (Gondal et al., 2019). We can see that the proposed two datasets – *Faclor3D* and *Isaac3D* both have much larger resolutions than previous datasets, together with the maximum number of factors. Furthermore, in terms of photorealism, both datasets are rendered based on a complex 3D scene, in particular with texturing in Isaac3D.

Intuitively, low interpolation variance $\beta_i(\phi)$ means that changing the factor $c_i$ does not affect much the generated images, which further implies the factor $c_i$ should not be considered as fine-grained, since FC-StyleGAN with downscaled resolution $\phi$ cannot control it any more.

## 4 EXPERIMENTS

In this section, we first introduce two new datasets – Isaac3D and Falcor3D. We then evaluate the performance of AC-StyleGAN and FC-StyleGAN on both datasets, focusing on three aspects: disentanglement quality, semantic correctness and image quality. Quantitatively, we use *Frechet Inception Distance (FID)* (Heusel et al., 2017) to measure image quality, *reconstruction error* (i.e., $l_{\text{rec}} = \|\hat{c}^{(r)} - c^{(r)}\|$) to measure semantic correctness, and *MIG* (Chen et al., 2018) to measure disentanglement quality. Qualitatively, we apply the latent traversals to visually evaluate all three aspects as well. Due to space limitations, we only show results on Isaac3D in this section; results on Falcor3D are quite similar to those on Isaac3D, and are available in the appendix.

### 4.1 CREATION OF HIGH-QUALITY DISENTANGLEMENT DATASETS

Current disentanglement datasets, such as 3D Chairs (Aubry et al., 2014), dSprites (Matthey et al., 2017) and MPI3D (Gondal et al., 2019), are of low resolution and mostly lack of photorealism. It makes them not suitable as disentanglement benchmarks in the high-resolution image domain. To this end, we propose two new datasets – *Falcor3D* and *Isaac3D*, which possess much higher resolution, better photorealism and richer factors of variations, as shown in Table 1.

**Falcor3D**   In the Falcor dataset, there are in total 233,280 images and each has a resolution of 1024x1024. This dataset is based on the 3D scene of a living room, where we can move the camera positions and change the lighting conditions. Each image is paired with a ground-truth meta code, consisting of 7 factors of variation: lighting intensity (5), lighting $x$-dir (6), lighting $y$-dir (6), lighting $z$-dir (6), camera $x$-pos (6), camera $y$-pos (6), and camera $z$-pos (6). Note that the number $m$ behind each factor represents that the factor has $m$ possible values, uniformly sampled in the normalized range of variations $[0, 1]$. To interpret this, for example, "lighting $x$-dir (6)" represents the lighting direction moving along the $x$-axis and "camera $z$-pos (6)" represents the camera position moving along the $z$-axis. Also, both factors have 6 values uniformly sampled from $[0, 1]$.

**Isaac3D**   In the Isaac3D dataset, there are in total 737,280 images and each has a resolution of 512x512. This dataset is based on the 3D scene of a kitchen, where we can also move the camera positions and vary the lighting conditions. To further increase the number of variations, we put a robotic arm inside, attached with an object. The robotic arm has two degrees of freedom: $x$-movement (or horizontal rotation) and $y$-movement (or vertical rotation). The attached object could change its shape, scale and color. To make the rendered images more photorealistic, each object in the 3D scene has been provided with proper textures. Finally, each image is paired with a ground-truth meta code, consisting of 9 factors of variation: lighting intensity (4), lighting $y$-dir (6), object color (4), wall color (4), object shape (3), object scale (4), camera height (4), robot $x$-movement (8), and robot $y$-movement (5). Similarly, the number $m$ behind each factor represents that the factor has $m$ possible values, uniformly sampled in the normalized range of variations $[0, 1]$.

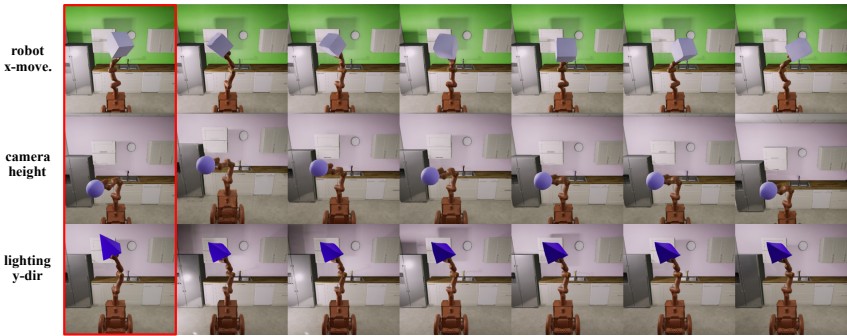

Figure 2: Latent traversal of AC-StyleGAN with full supervision on Isaac3D. For illustration, we only show three factors: (robot $x$-movement, camera height, lighting $y$-dir). Please see the appendix for latent traversal of all the factors. Images in the first column (marked by red box) are randomly sampled real images with the ground-truth and the rest images in each row are their interpolations, respectively, by uniformly varying the given factor from 0 to 1. Unless otherwise stated, this setting applies to all the latent traversal results below.

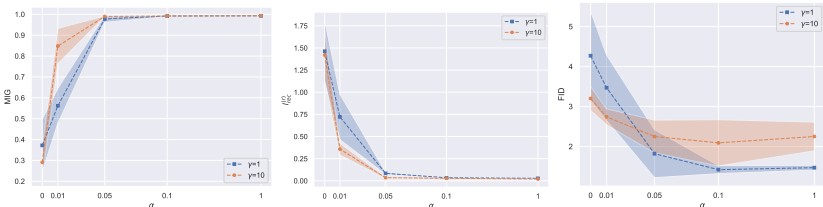

Figure 3: Quantitative metrics – MIG, $l_{rec}$ and FID vary with the supervision coefficient $\alpha \in \{0.0, 0.01, 0.05, 0.1, 1.0\}$ and the disentanglement coefficient $\gamma \in \{1, 10\}$ in AC-StyleGAN on Isaac3D. For MIG, the higher is better, while for $l_{rec}$ and FID, the lower is better.

Please see the appendix for more detailed descriptions of the two datasets.

## 4.2 EVALUATION OF AC-STYLEGAN

**Latent traversal of AC-StyleGAN.** We first show the latent traversal results of AC-StyleGAN on Isaac3D in Figure 2, where we apply full supervision from meta code of all the factors. For illustration, here we only show three factors of variation: (robot $x$-movement, camera height, lighting $y$-dir). Please see the appendix for latent traversal of all the factors. Images in the first column (marked by red box) are randomly sampled real images and the rest images in each row are their interpolations, respectively, by uniformly varying each factor from 0 to 1. Unless otherwise stated, this setting applies to all the latent traversal results below. As we can see, each factor in the interpolated images changes smoothly without affecting other factors. Taking "lighting $y$-dir" as an example, only the direction of the point light gradually moves up from left to right in the bottom row, which can be evidenced by the fact that the light spot moves up from the refrigerator to the wall and the shadow of the cabinet on the wall moves down simultaneously. More importantly, all the interpolated images visually look the same with the corresponding real images in the first column except for the interpolated factors, and the image quality does not degrade over interpolation. Therefore, the latent traversal results demonstrate three good properties of AC-StyleGAN with supervision: high disentanglement quality and good semantic correctness and high generation quality.

**Semi-supervised learning.** How does AC-StyleGAN behave when given fewer labels? To this end, we vary the supervision coefficient $\alpha$ in Eq. (2) to show the impact of supervision on AC-StyleGAN and the results with $\gamma \in \{1, 10\}$ on Isaac3D are in Figure 3. First, we can see with limited supervision ($\alpha \geq 0.1$), the difference between $\gamma = 1$ and 10 is quite small. Also, both values of MIG and $l_{rec}$ almost reach their own optimal ones, along with a lower FID. This quantitatively supports the above latent traversal results in Figure 2. As expected, the disentanglement quality and semantic correctness gradually get worse when less labelled data is used. Interestingly, the gap between $\gamma = 1$ and 10 first increases and then decreases as $\alpha$ decreases from 0.5 to 0, with the largest

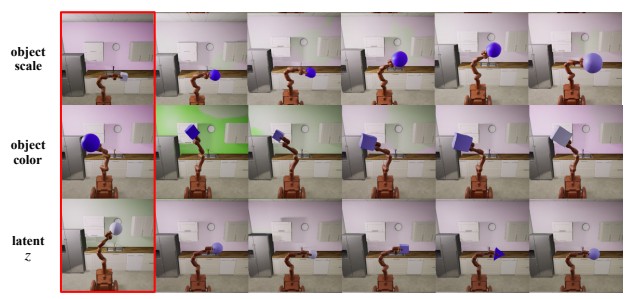
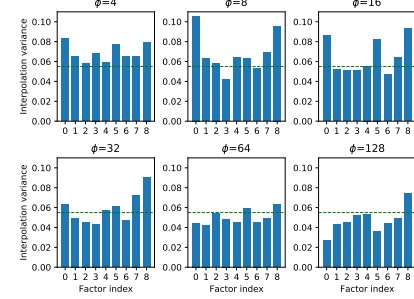

(a) Supervision from a subset of factors      (b) Identifying fine-grained factors

Figure 4: (a) Latent traversal of AC-StyleGAN on Isaac3D, where we only control a subset of factors: (robot $x$-movement, robot $y$-movement, object scale, lighting $y$-dir, object color) and here two of them are shown. Please see the appendix for all five considered factors. (b) Interpolation variances of each factor in FC-StyleGAN with different downscaled resolutions $\phi$ on Isaac3D, where the meta code of all factors is used as its input. Empirically, the factor with its interpolation variance below the threshold (marked by the green line) cannot be controlled by FC-StyleGAN with the given $\phi$.

one at $\alpha = 0.01$. It means that properly increasing $\gamma$ is the most beneficial for disentanglement in the case where only a very limited number of labelled data is provided. As we can see, by only using 1% of the labelled data ($\alpha = 0.01$) and setting $\gamma = 10$, the disentanglement quality and semantic correctness are still close to those with full supervision ($\alpha = 1$), and improve over the unsupervised baseline ($\alpha = 0$) by a significant margin. Please see the appendix for the latent traversal results of AC-StyleGAN with $\alpha = 0.01$ and $\gamma = 10$. It means adding a very small amount of labelled data into the training dataset could benefit much the disentanglement learning. Finally, in terms of image quality, we observe that larger $\alpha$ tends to result in a lower FID score, which means better disentanglement is not always obtained by sacrificing the generation quality.

**Comparisons with different baselines.** Current start-of-the-art approaches for *generic* unsupervised disentanglement learning are mostly VAE-based models, such as $\beta$-VAE (Higgins et al., 2017), FactorVAE (Kim & Mnih, 2018) and $\beta$-TCVAE (Chen et al., 2018). To this end, we use the unsupervised version of AC-StyleGAN ($\alpha = 0$) to compare with VAE-based models on (i) the dSprites dataset and (ii) the Isaac3D dsataset, respectively. Note that for evaluations on Isaac3D, we first downscale its resolution to 128x128 because VAEs have difficulties in generating higher-resolution images. We can see from Table 2 that the unsupervised AC-StyleGAN consistently outperforms all the start-of-the-art disentangled VAEs on both datasets. Particularly on the downscaled Isaac3D dataset, disentangled VAEs have much higher FID scores (FID > 100), indicating their poor image generation quality. Please see Appendix A.5 for their randomly sampled images. Interestingly, the disentangled VAEs also fail to capture all the variations in the Isaac3D dataset, supporting the claim that our datasets provide larger variation of factors for disentanglement learning.

Previous work on the semi-supervised learning for classification has shown that a supervised baseline with only the labelled data could be very competitive (Oliver et al., 2018; Zhai et al., 2019). Hence, to demonstrate the effectiveness of actual semi-supervised components for disentanglement, we also compare our semi-supervised method with the supervised baseline, in which we remove the unsupervised disentanglement term in the discriminator loss function of Eq. (2). The results are shown in Table 3. We can see that when only 1% or 5% of lablled data is available, semi-supervised AC-StyleGAN consistently outperforms the supervised baseline in terms of all three metrics: FID, MIG and $l_{rec}$. Moreover, the advantages of using the unsupervised disentanglement term become larger when we have fewer labelled data.

**Latent traversal on 256x256 CelebA.** To show the performance of AC-StyleGAN on the real-world data, we also provide the qualitative results on CelebA with resolution 256x256 (Liu et al., 2015). In this experiment, the meta code in AC-StyleGAN has a length of 40 to capture all the attributes in the dataset. Hence, this is more challenging than the previous works on CelebA in terms of controlling all the attributes in a higher resolution (Chen et al., 2016; Higgins et al., 2017; Nguyen-Phuoc et al., 2019). Note that since CelebA only has binary attributes, which commonly-used disentanglement metrics including MIG and FactorVAE score are not applicable to, we focus

| Methods | FactorVAE Score | MIG |
|---|---|---|
| $\beta$-VAE | $0.731 \pm 0.020$ | $0.177 \pm 0.031$ |
| FactorVAE | $0.757 \pm 0.107$ | $0.157 \pm 0.051$ |
| $\beta$-TCVAE | $0.783 \pm 0.057$ | $0.198 \pm 0.019$ |
| Ours ($\gamma = 1$) | $0.760 \pm 0.050$ | $0.213 \pm 0.090$ |
| Ours ($\gamma = 10$) | $\mathbf{0.840 \pm 0.090}$ | $\mathbf{0.290 \pm 0.098}$ |

(a) dSprites with resolution 64x64

| Methods | FID | MIG |
|---|---|---|
| $\beta$-VAE | $122.6 \pm 2.0$ | $0.231 \pm 0.068$ |
| FactorVAE | $305.8 \pm 142.1$ | $0.245 \pm 0.034$ |
| $\beta$-TCVAE | $155.4 \pm 13.6$ | $0.216 \pm 0.074$ |
| Ours ($\gamma = 1$) | $4.27 \pm 1.12$ | $\mathbf{0.372 \pm 0.126}$ |
| Ours ($\gamma = 10$) | $\mathbf{3.20 \pm 0.30}$ | $0.291 \pm 0.077$ |

(b) Isaac3D with resolution 128x128

Table 2: Comparison of unsupervised AC-StyleGAN and state-of-the-art VAE-based models: $\beta$-VAE, FactorVAE and $\beta$-TCVAE on the dSprites and downscaled Isaac3D datasets. Note that the scores of VAE-based models are obtained based on the implementation in Locatello et al. (2019a), where we set $\beta = 6$ for $\beta$-VAE, $\gamma = 30$ for FactorVAE and $\beta = 8$ for $\beta$-TCVAE after a grid search over different hyperparameters.

| | Methods | FID | MIG | $l_{rec}$ |
|---|---|---|---|---|
| 5% | Sup. baseline | $2.80 \pm 0.22$ | $0.963 \pm 0.007$ | $0.131 \pm 0.013$ |
| | Ours | $\mathbf{2.25 \pm 0.41}$ | $\mathbf{0.990 \pm 0.002}$ | $\mathbf{0.035 \pm 0.010}$ |
| 1% | Sup. baseline | $3.58 \pm 0.20$ | $0.766 \pm 0.042$ | $0.358 \pm 0.067$ |
| | Ours | $\mathbf{2.74 \pm 0.20}$ | $\mathbf{0.848 \pm 0.085}$ | $\mathbf{0.231 \pm 0.157}$ |

Table 3: Comparison of semi-supervised AC-StyleGAN with the supervised baseline, where we use $\gamma = 10$ and consider two cases of using 5% and 1% of labelled data. In the supervised baseline, we remove the unsupervised disentanglement term in the discriminator loss function of Eq. (2).

on the qualitative results of latent traversal. The results are shown in Figure 5. We can see that AC-styleGAN can smoothly interpolate in the latent space of face attributes and also maintain high image quality. More interesting interpolations can be seen in Appendix A.6, where some attributes are more difficult to change during latent traversal, presumably due to the fact that these attributes have low frequency in the CelebA dataset, such as Mustache on female faces.

**Only controlling a subset of style factors.** Instead of trying to control all the factors in a dataset, practically speaking we may only be interested in a subset of factors in which case the remaining factors will be considered as *random nuisances* captured by the latent $z$. On a first look, it might seem intuitive that the fewer factors of variation we try to control, the better the disentanglement quality we should achieve. However, the latent traversal results of AC-StyleGAN( Figure 4a) show instead that both disentanglement quality and semantic correctness degrade. For example, the camera height changes when interpolating the object scale, and the wall color also varies when interpolating the object scale. Furthermore, the latent $z$, which is expected to capture the remaining factors: (object shape, camera height, lighting intensity, wall color), can barely change the camera height and wall color (see the appendix for results of other factors). To explain this, we argue that since factors are highly correlated with each other, learning to control one may help a lot in learning to control the others. Without observations on other nuisance factors, it may be very difficult to disentangle the impact of the observed factor with unobserved covariates, especially in the high-dimensional space.

### 4.3 EVALUATION OF FC-STYLEGAN

**Identifying fine-grained factors.** As we have discussed, it is a crucial for FC-StyleGAN to quantitatively identify fine-grained factors. Here we calculate the interpolation variance of each factor according to Eq. (3) by setting $N = 100, S = 10$. Figure 4b shows the interpolation variance of each factor in FC-StyleGAN with downscaled resolution $\phi$ varying from 4 to 128 on Isaac3D. To decide the threshold $\beta_0$, we visually check the latent traversal results and find that the interpolations over a factor $c_i$ almost stay the same if $\beta_i(\phi) < 0.055$, which means $\beta_0 = 0.055$. Please see the appendix for the latent traversal results. We can see that as $\phi$ gets larger, more factors have an interpolation variance below the threshold $\beta_0$, meaning more coarse-grained factors appear. For instance, there are only two coarse-grained factors at $\phi = 8$: camera height and lighting direction, and two more factors become coarse-grained factors at $\phi = 16$: robot $x$-movement and robot $y$-movement, and so on. Interestingly, wall color is always fine-grained even at $\phi = 128$ while object color has already become coarse-grained at $\phi = 64$, although both of them are about colors.

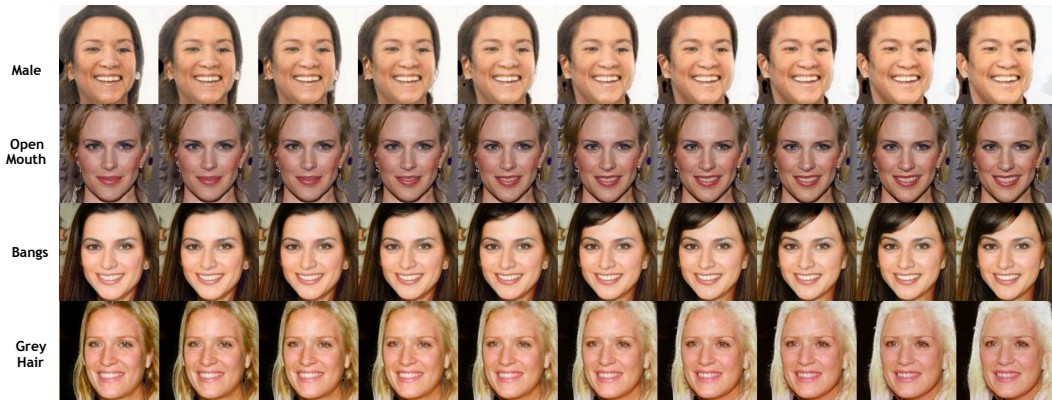

Figure 5: Latent traversal of AC-StyleGAN on CelebA with resolution 256x256, where we control all 40 binary attributes at the same time and show four representative ones: Male, Open Mouse, Bangs, Grey Hair.

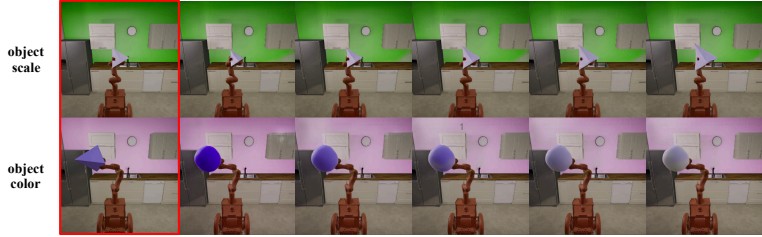

Figure 6: Latent traversal of FC-StyleGAN with downscaled resolution $\phi = 16$ on Isaac3D, where we only control a subset of fine-grained factors: (object scale, lighting intensity, object color, wall color) and here two of them are shown. Please see the appendix for all the four factors.

**Latent traversal of FC-StyleGAN.** Based on the above observations, we can obtain the latent traversal results of FC-StyleGAN on fine-grained factors, which are shown in Figure 6. Specifically, we train FC-StyleGAN with downscaled resolution $\phi = 16$ on Isaac3D. Since Figure 4b indicates that fine-grained factors at $\phi = 16$ are of index $(0, 4, 5, 7, 8)$, we use its subset: (object scale, lighting intensity, object color, wall color) as the input while leaving the object shape together with all the coarse-grained factors as random nuisances. We can see each considered factor in the interpolated images changes smoothly without affecting other factors, which implies good disentanglement quality. Because FC-StyleGAN is only applicable for fine-grained control, we can observe that, on the one hand, the coarse-grained factors of the interpolated images always stay the same with the corresponding input image, even though they are random nuisances. On the other hand, the object shape, as the only one fine-grained random nuisance, could be different from the corresponding input image with some probability. In Figure 6, for example, the object shape stays the same with the input image when interpolating the object scale, while it becomes different from the input image when interpolating the object color. More importantly, compared with the results of AC-StyleGAN in Figure 4a, FC-StyleGAN has much better disentanglement quality, when it comes to fine-grained control with supervision from a subset of factors.

**Impact of instance normalization.** Now we want to show that without instance normalization in FC-StyleGAN, the input image can interference the control of fine-grained code over the output fine styles. The results of FC-StyleGAN with and without instance normalization are shown in Figure 7a, where both are trained with downscaled resolution $\phi = 32$ on Isaac3D, and only a subset of fine-grained factors (lighting intensity, object color, wall color) is used as supervision. For illustration, Figure 7a only shows the the wall color for comparison. Please see the appendix for all three factors. We can see that without instance normalization, latent traversal in FC-StyleGAN cannot change the wall color any more, which instead stays the same with the input image. This confirms our previous hypothesis that the instance normalization plays an important role in alleviating the impact of the input image in controlling fine-grained features over generation. Also, the quantitative results in

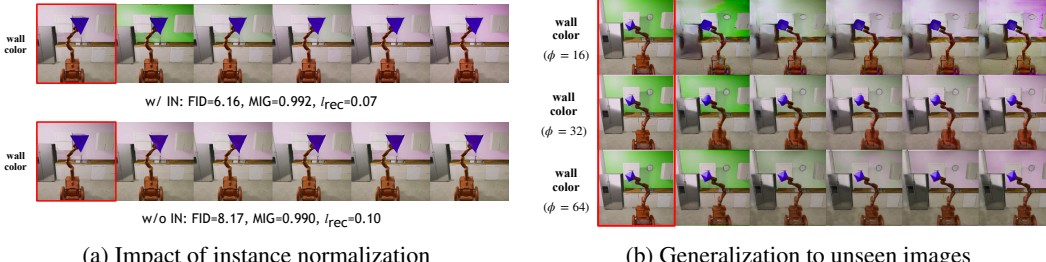

(a) Impact of instance normalization          (b) Generalization to unseen images

Figure 7: (a) Comparison of FC-StyleGAN with instance normalization (top row) and without instance normalization (bottom row). Both are trained with downscaled resolution $\phi = 32$ on Isaac3D, where we only disentangle a subset of fine factors (lighting intensity, object color, wall color) and here only one are shown (see the appendix for all three considered factors). (b) Generalization of FC-StyleGAN by varying the downscaled resolution $\phi$ and interpolating the wall color. In the test image, we shift the position of the robot arm to the right-hand side, which is also attached with an unseen object (i.e., octahedron).

Figure 7a further confirm the advantages of using instance normalization. In particular, the fact that the reconstruction error $l_{\text{rec}}$ becomes larger without instance normalization implies that the strong interference of the input image also degrades the prediction of fine-grained code in the discriminator.

**Generalization to unseen images.**   As an image-to-image model, it is natural to ask how FC-StyleGAN generalizes to unseen images. To test its generalization ability, the novel test images are provided as follows: i) we shift the robot position to the right hand side of the camera (instead of standing right in the middle for the training dataset), and ii) we also attach the robot arm with an unseen object. Figure 7b shows the results of interpolating the wall color of the same test image with different downscaled resolutions $\phi \in \{16, 32, 64\}$. Please see the appendix for results of other test images and different fine-grained factors. As we can see, the wall color keeps changing during its interpolations in each case without affecting other factors, implying good disentanglement quality. Furthermore, the interpolated images in each case maintain the new robot position and particularly maintain new object shape (i.e., octahedron) in the case of $\phi = 64$. The reason the new object shape is only maintained at $\phi = 64$ is because that the object shape, together with the robot position, is not fine-grained at $\phi = 64$ any more. Therefore, it demonstrates that the disentanglement learning of FC-StyleGAN can generalize well to unseen novel test images. Finally, Figure 7b shows that the generalization results get better in terms of image quality as we increase the downscaled resolution $\phi$. As we know, with the larger value of $\phi$, we can control fewer fine-grained factors. Therefore, there exists a trade-off between the generalization and controllability in FC-StyleGAN. We leave the investigation into how to improve this trade-off in the future work.

## 5   CONCLUSIONS

In this work, we developed AC-StyleGAN, by combining both advantages of AC-GAN and Style-GAN, to demonstrate that it is possible to perfectly disentangle and control different factors of variation in the high-resolution image domain with sufficient supervision. In particular, with only 1% of labelled data, the disentanglement quality is very close to the fully supervised case and outperforms the fully unsupervised case by a significant margin, emphasizing the importance of (weak) supervision. To address the performance degradation of AC-StyleGAN in cases where we observe/control only a subset of factors, we proposed FC-StyleGAN, a new image-to-image model for semantically manipulating fine-grained factors of a given high-fidelity image. We demonstrated the importance of instance normalization in FC-StyleGAN, and also showed its good generalization ability to unseen images. Finally, we proposed two new datasets – Falcor3D and Isaac3D with higher resolution, better photorealism and richer factors of variation, compared to current disentanglement datasets. We believe that our proposed models, new datasets, along with the useful insights we have gleaned, can benefit the further development of disentangled GANs for the controllable generation of high-fidelity images. For the future work, we would like to explore how to control coarse-grained factors only in high-resolution images while leaving fine-grained factors as random nuisances.

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

# A APPENDIX

## A.1 MORE DETAILS OF THE PROPOSED TWO DATASETS

### A.1.1 EXAMPLES IN THE ISAAC3D DATASET

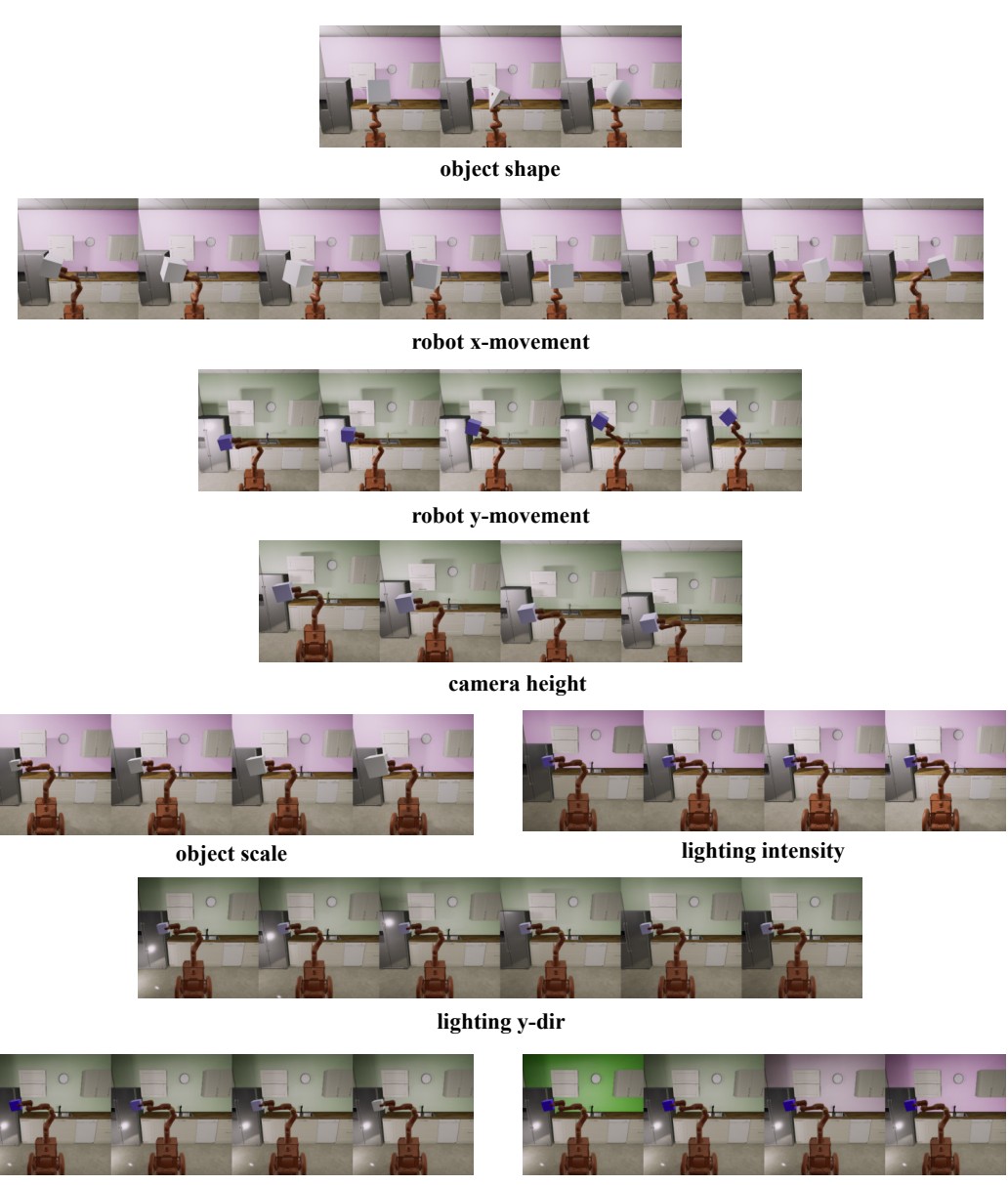

Figure 8: Examples in the Isaac3D dataset where we vary each factor of variation individually.

### A.1.2 EXAMPLES IN THE FALCOR3D DATASET

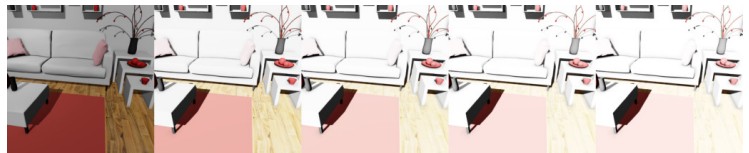

**lighting intensity**

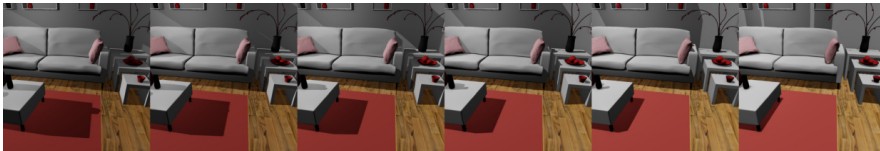

**lighting x-dir**

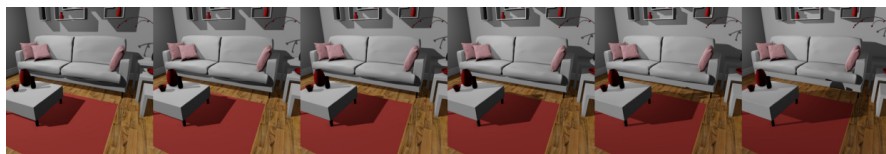

**lighting y-dir**

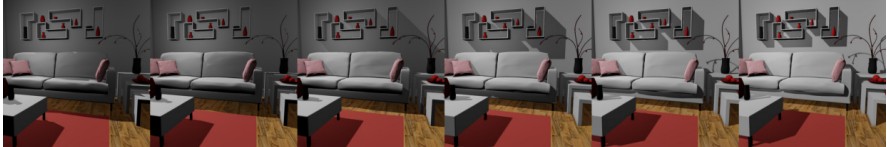

**lighting z-dir**

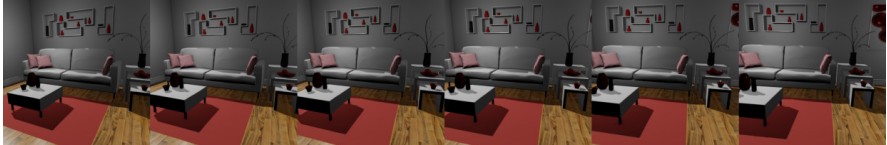

**camera x-pos**

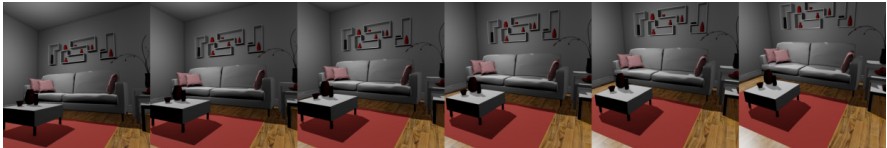

**camera y-pos**

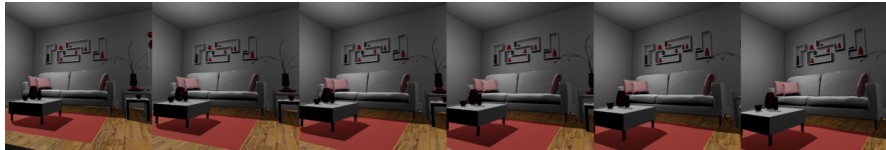

**camera z-pos**

Figure 9: Examples in the Falcor3D dataset where we vary each factor of variation individually.

## A.2 IMAGE RECONSTRUCTIONS WITH THE SAME META CODE IN AC-STYLEGAN

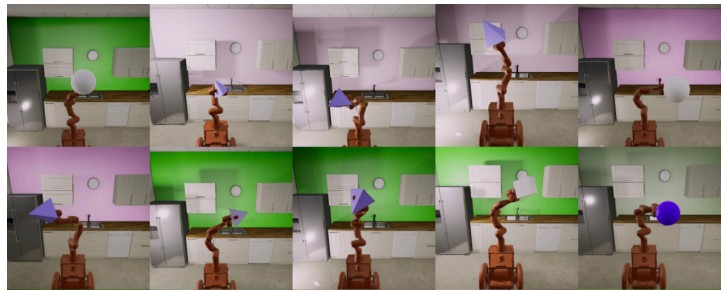

**real images**

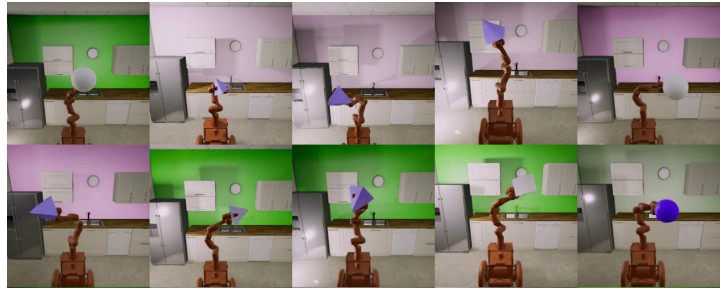

**fake images**

(a) Image reconstruction on random samples of Isaac3D

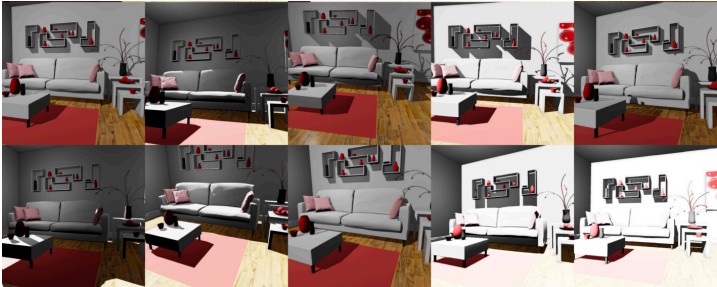

**real images**

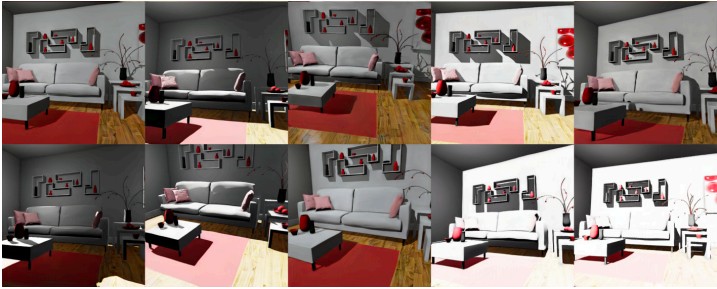

**fake images**

(b) Image reconstruction on random samples of Falcor3D

Figure 10: Image reconstruction on random samples of Isaac3D and Falcor3D in AC-StyleGAN with full supervision, where we can see that the generated fake images match well with real images by using the same meta code as the generator input, confirming the semantic correctness of the model.

## A.3 MORE RESULTS ON LATENT TRAVERSAL OF AC-STYLEGAN

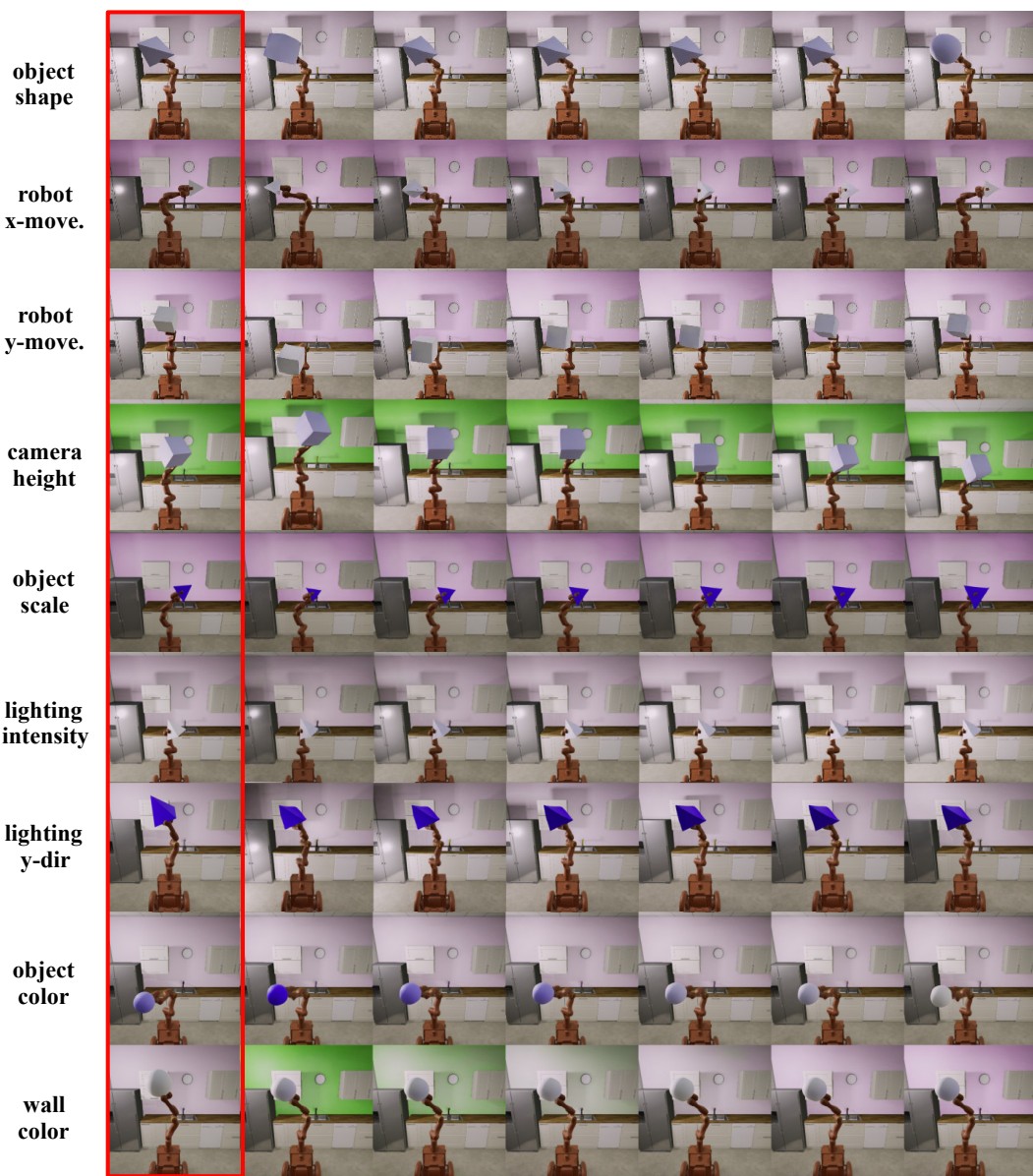

Figure 11: Latent traversal results of AC-StyleGAN with full supervision on Isaac3D for all the factors. Images in the first column (marked by red box) are randomly sampled real images of resolution 512x512 and the rest images in each row are their interpolations, respectively, by uniformly varying the given factor from 0 to 1. We can see that each factor changes smoothly during its interpolation without affecting other factors, and the interpolated images in each row visually look almost the same with their input image except the considered varying factor. Also, the image quality does not get worse during the interpolations.

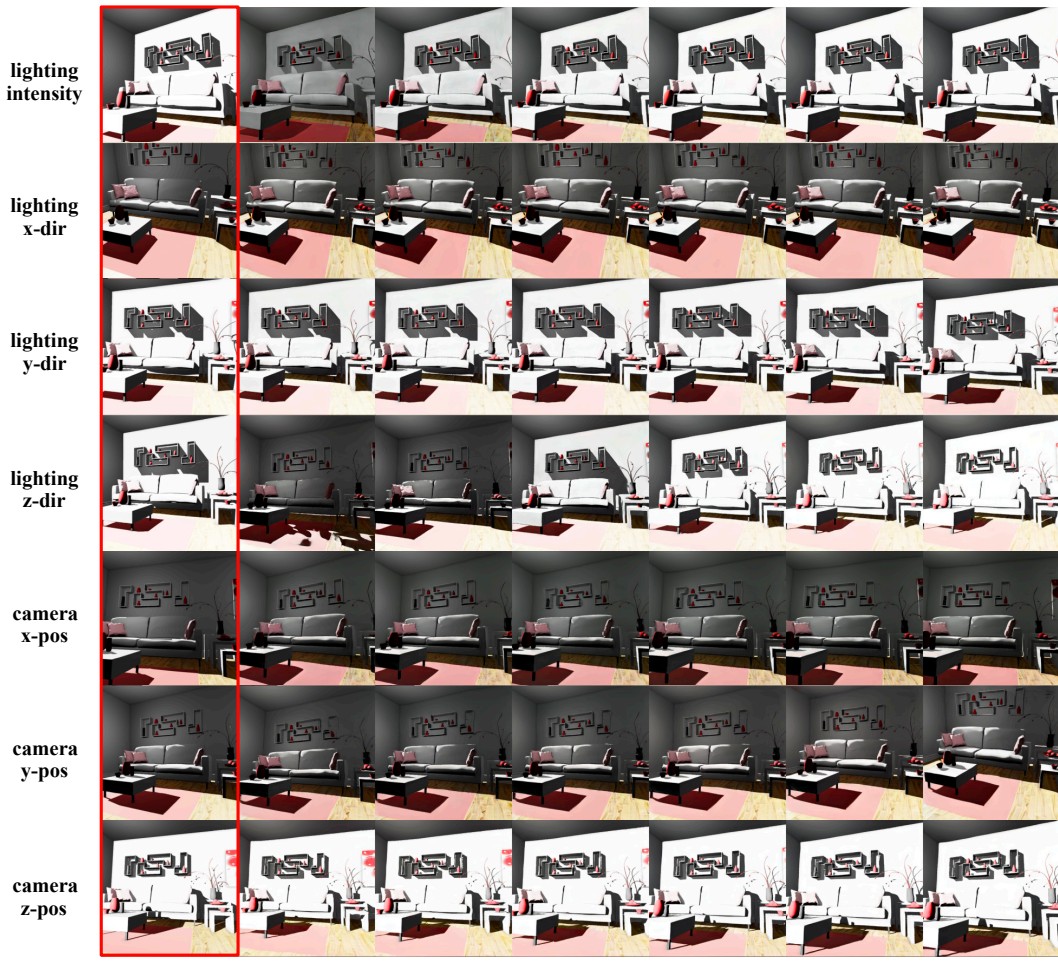

Figure 12: Latent traversal results of AC-StyleGAN with full supervision on Falcor3D for all the factors. Images in the first column (marked by red box) are randomly sampled real images of resolution 1024x1024 and the rest images in each row are their interpolations, respectively, by uniformly varying the given factor from 0 to 1. We can see that each factor changes smoothly during its interpolation without affecting other factors, and the interpolated images in each row visually look very similar to their input image except the considered varying factor. Also, the image quality does not get worse during the interpolations.

## A.4 Latent traversal results AC-StyleGAN with 1% labelled data

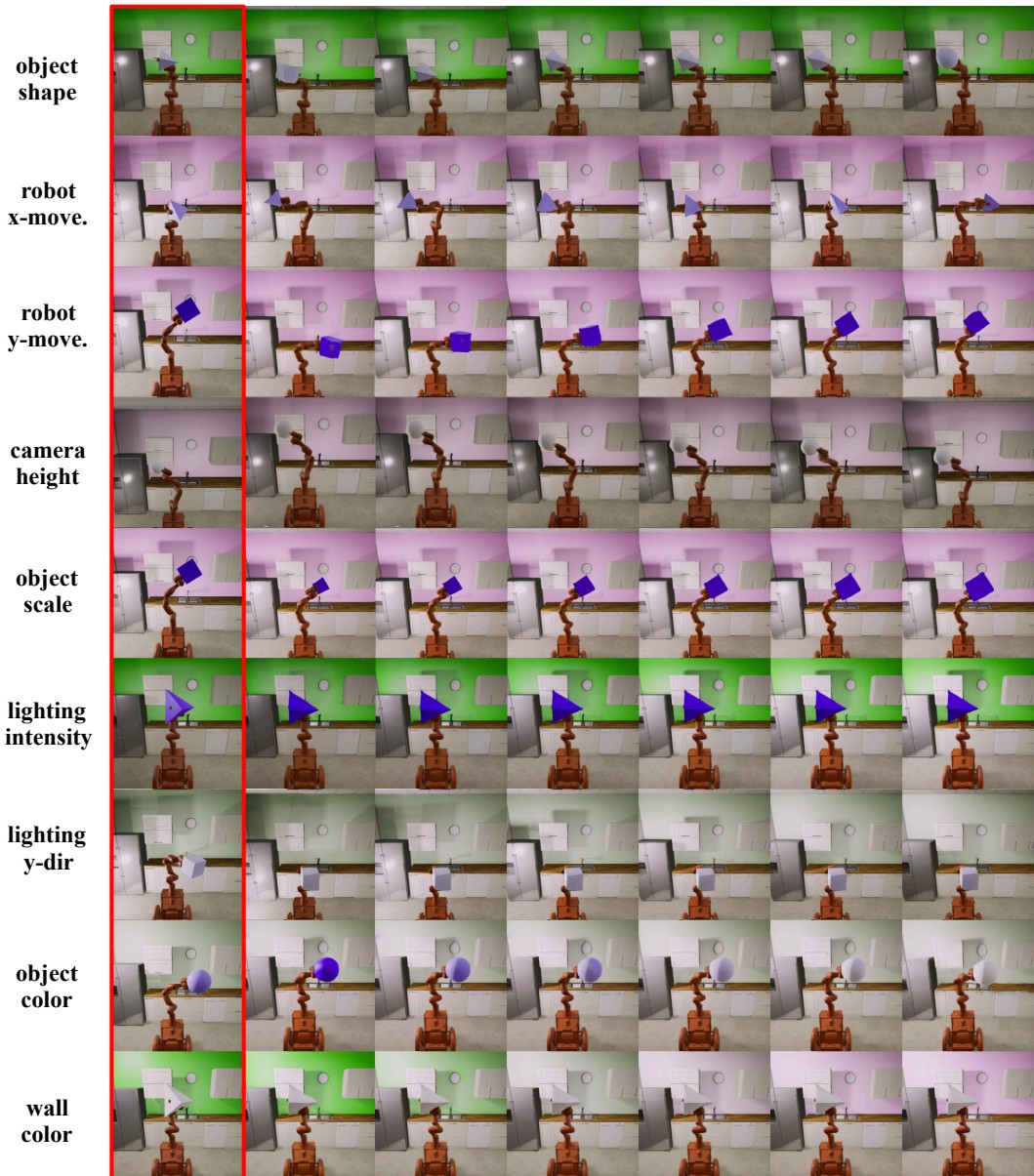

Figure 13: Latent traversal results of AC-StyleGAN with semi-supervision ($\alpha = 0.01$) on Isaac3D for all the factors. Images in the first column (marked by red box) are randomly sampled real images of resolution 512x512 and the rest images in each row are their interpolations, respectively, by uniformly varying the given factor from 0 to 1. We can see that in most cases, each factor changes smoothly during its interpolation without affecting other factors except the entanglement between the object shape and camera height. Also, except the considered varying factor, the interpolated images in each row visually look similar to their input image with small shifts sometimes. It implies a reasonably good disentanglement quality and semantic correctness in the AC-StyleGAN with 1% labelled data. Similarly, the image quality does not get worse during the interpolations.

## A.5 RANDOMLY SAMPLED IMAGES OF DISENTANGLED VAEs ON THE DOWNSCALED ISAAC3D DATASET

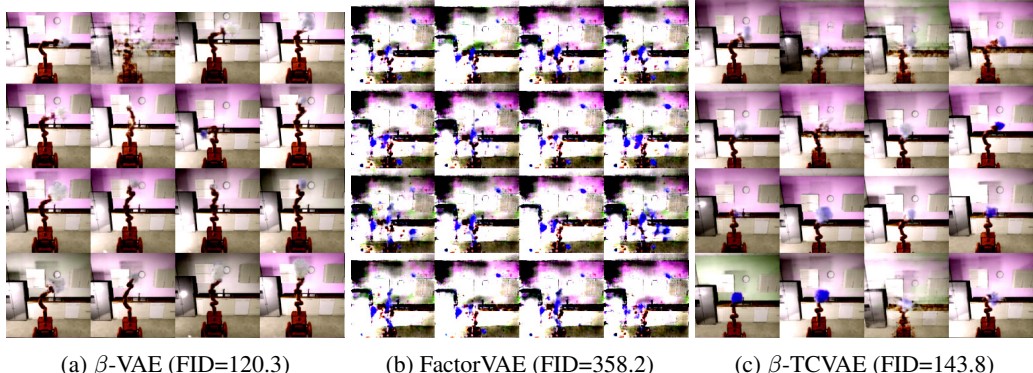

(a) $\beta$-VAE (FID=120.3)  (b) FactorVAE (FID=358.2)  (c) $\beta$-TCVAE (FID=143.8)

Figure 14: The randomly sampled images of disentangled VAEs on the Isaac3D dataset of resolution 128x128, where we use the same network architectures as in Locatello et al. (2019a). We can see that compared with AC-StyleGAN, (i) the generated images tend to be quite blurry and of low quality, (ii) the generated images fail to cover all the variations in the dataset. The results of disentangled VAEs also demonstrate that our proposed datasets serve as a new disentanglement challenge, in particular regarding the much higher resolution, and larger variation of factors.

## A.6 MORE LATENT TRAVERSAL RESULTS OF AC-STYLEGAN ON CELEBA

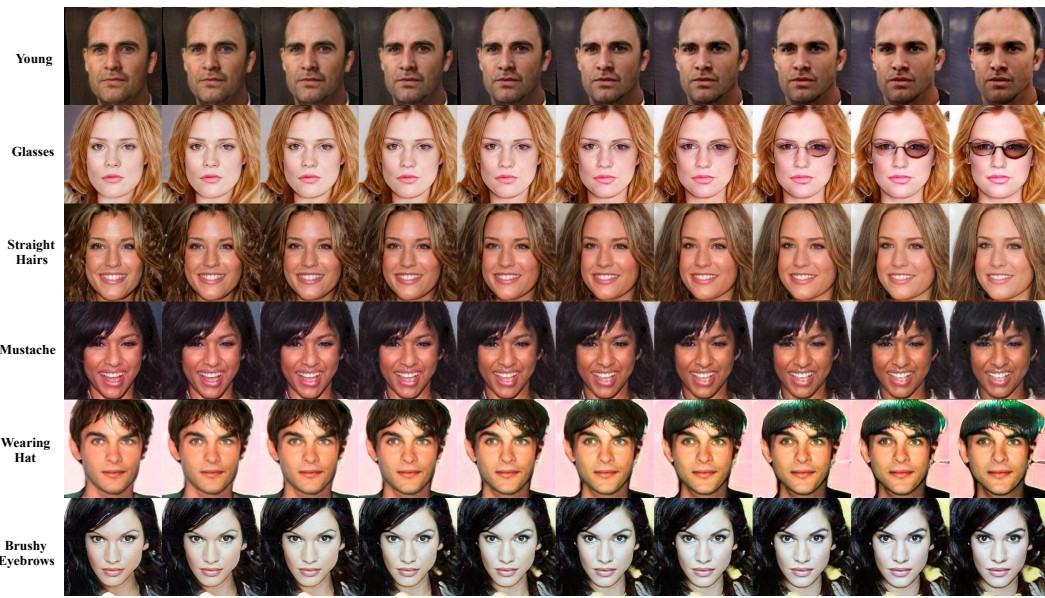

Figure 15: More Latent traversal results of AC-StyleGAN on CelebA with resolution 256x256, where we control all 40 binary attributes at the same time. We can see that AC-StyleGAN is capable of controlling most attributes. Some attributes are more difficult to change over interpolations, such Mustache on the female faces. We argue that this is because these attributes have low frequency in the CelebA dataset.

A.7    MORE RESULTS ON ONLY CONTROLLING A SUBSET OF FACTORS IN AC-STYLEGAN

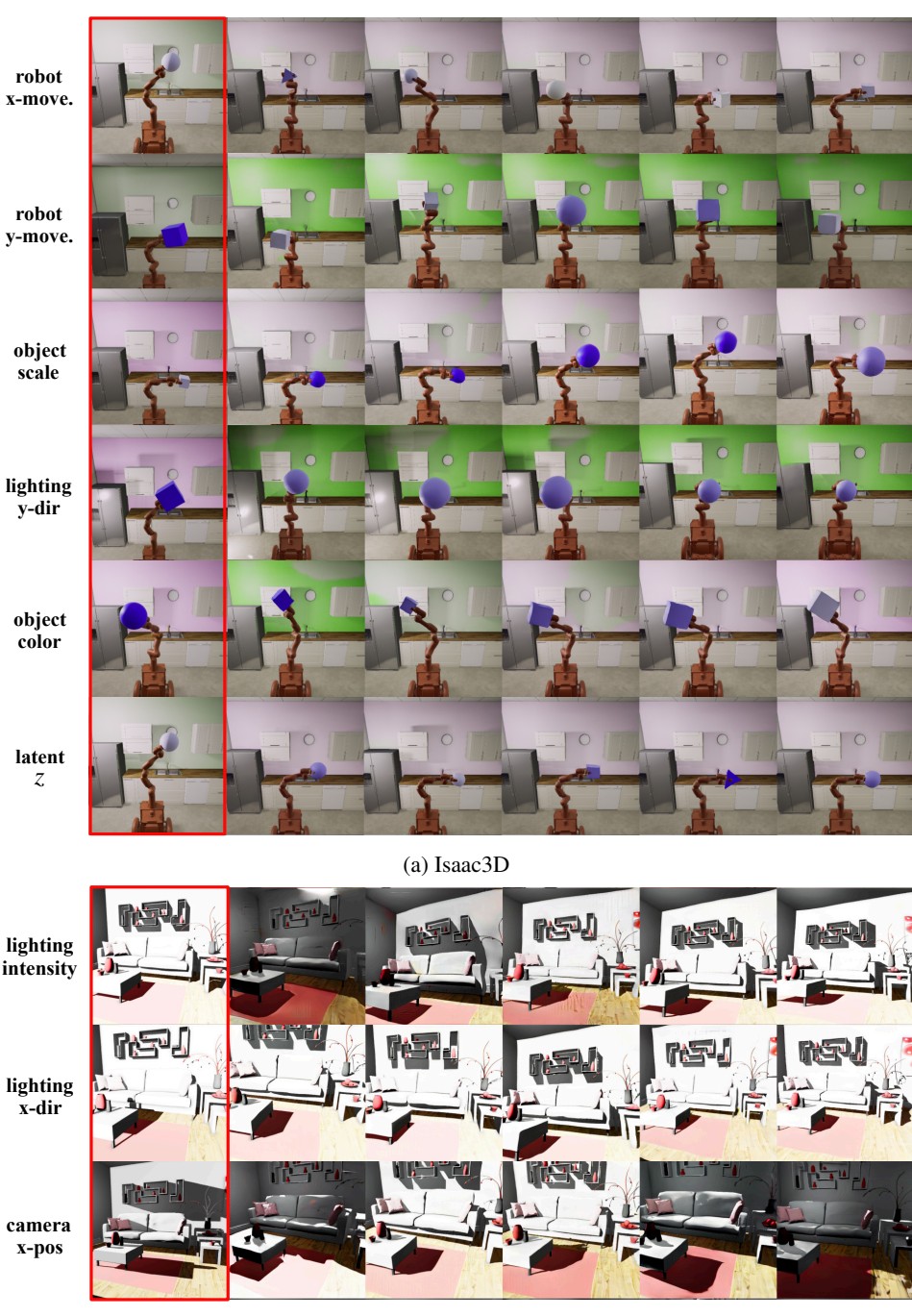

(a) Isaac3D

(b) Falcor3D

Figure 16: Latent traversal results of AC-StyleGAN for only controlling a subset of factors on a) Isaac3D: (robot $x$-movement, robot $y$-movement, object scale, lighting $y$-dir, object color), b) Falcor3D: (lighting intensity, lighting $x$-dir, camera $x$-pos). The other factors in each dataset will be considered as random nuisances, presumably captured by the latent $z$. We can see that the disentanglement quality and semantic correctness are both getting worse instead. For example, interpolating the lighting intensity of Falcor3D also changes the lighting directions and camera positions.

A.8    MORE RESULTS ON IDENTIFYING FINE-GRAINED FACTORS

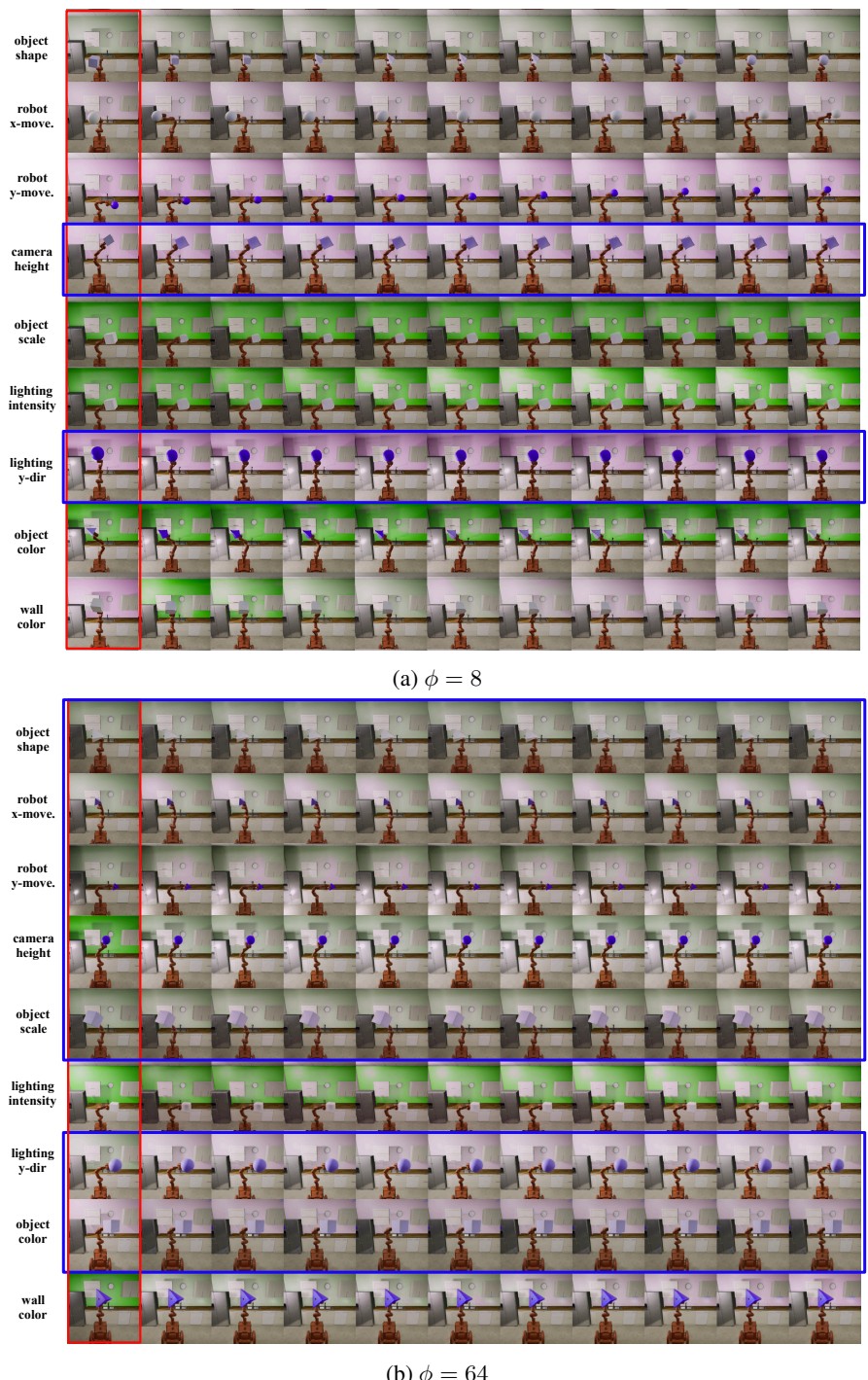

Figure 17: Latent traversal results of FC-StyleGAN with full supervision on Isaac3D for all the factors where we show different downscaled values $\phi \in \{8, 64\}$. The factors that cannot be changed by the interpolations (i.e., not fine-grained) are highlighted by blue boxes. For example, if $\phi = 8$, only the camera height and lighting $y$-dir are NOT fine-grained, while if $\phi = 64$, only the lighting intensity and wall color are fine-grained. Note that the results are consistent to those in Figure 4b.

## A.9 MORE RESULTS ON THE LATENT TRAVERSAL OF FC-STYLEGAN

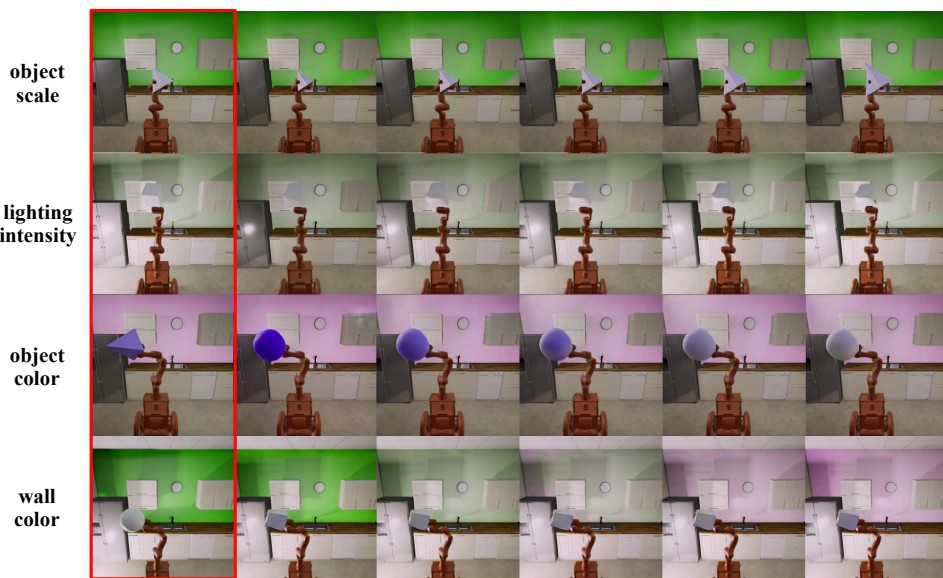

Figure 18: Latent traversal of FC-StyleGAN with full supervision on Isaac3D for a subset of fine-grained factors: (object scale, lighting intensity, object color, wall color). We can see that each factor changes smoothly during its interpolation without affecting other factors, and the interpolated images in each row visually look almost the same with their input image except the considered varying factor and another fine-grained factor: object shape (as a random nuisance). Also, the image quality does not get worse during the interpolations.

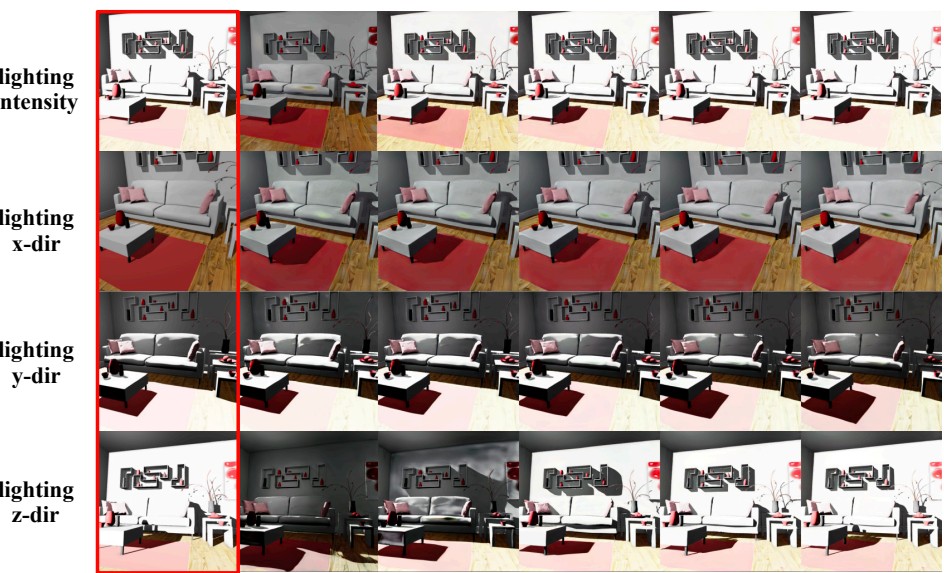

Figure 19: Latent traversal of FC-StyleGAN with full supervision on Falcor3D for all fine-grained factors: (lighting intensity, lighting $x$-dir, lighting $y$-dir, lighting $z$-dir). We can see that each factor changes smoothly during its interpolation without affecting other factors, and the interpolated images in each row visually look almost the same with their input image except the considered varying factor. Also, the image quality does not get worse during the interpolations.

A.10   More results on the impact of instance normalization in FC-StyleGAN

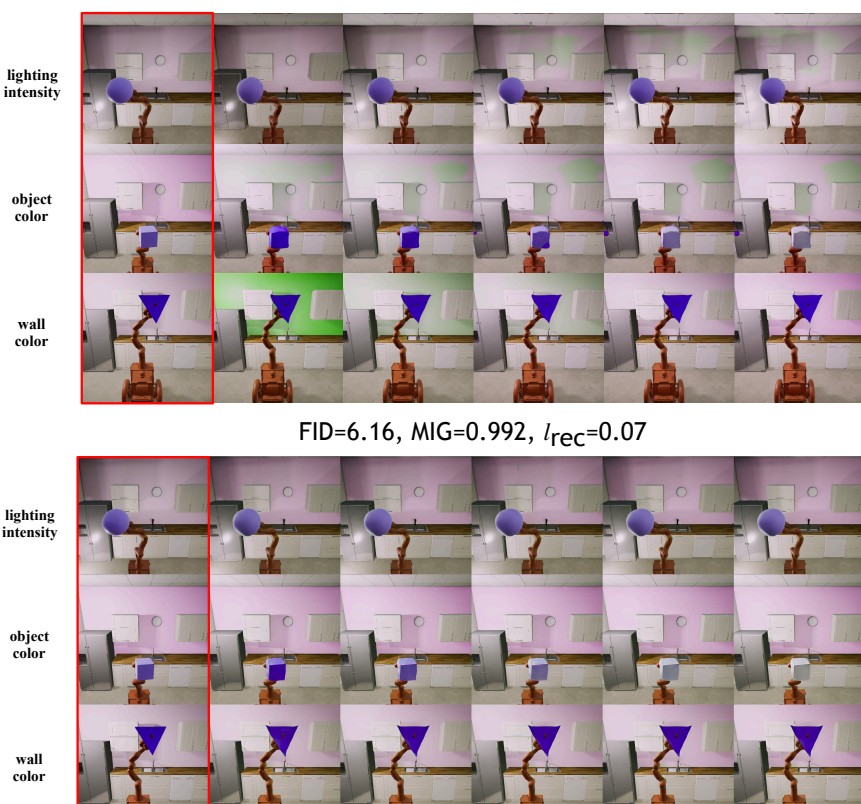

(a) Impact of instance normalization

Figure 20: Comparison of FC-StyleGAN with instance normalization (top row) and without instance normalization (bottom row). Both are trained with downscaled resolution $\phi = 32$ on Isaac3D, where we only disentangle a subset of fine factors (lighting intensity, object color, wall color). We can see that FC-StyleGAN with instance normalization can smoothly change each of the factors of variation over interpolation. However, FC-StyleGAN without instance normalization cannot change the lighting intensity and wall color at all.

## A.11    MORE RESULTS ON THE GENERALIZATION OF FC-STYLEGAN

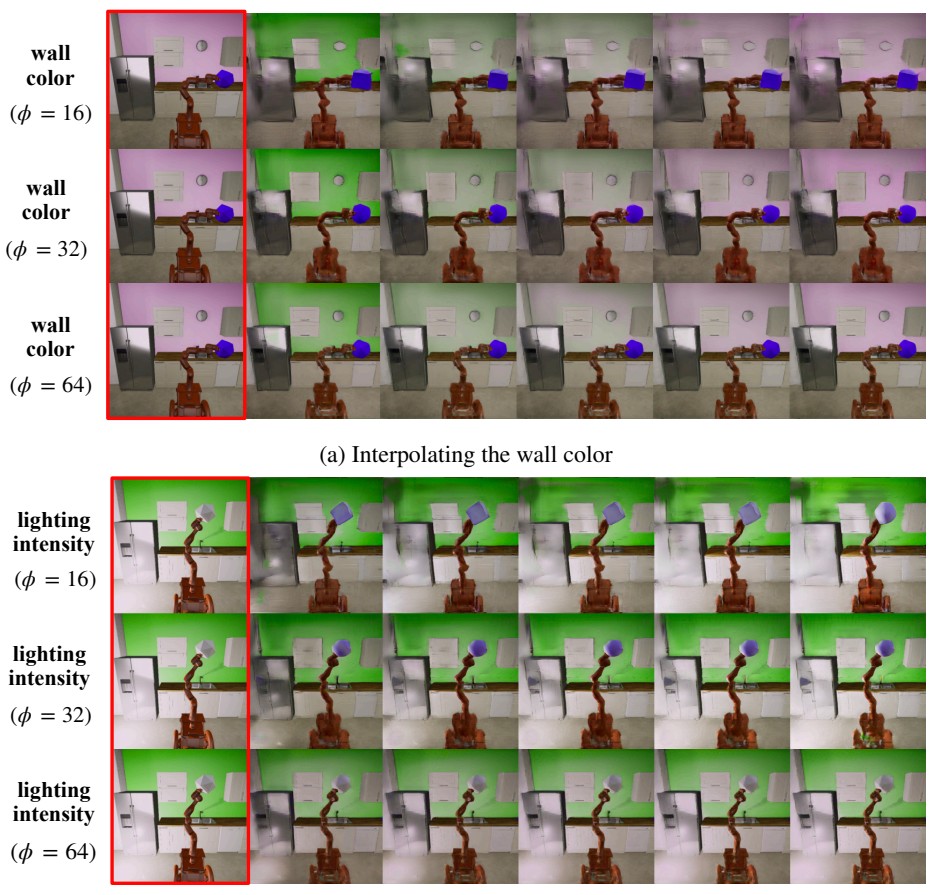

(a) Interpolating the wall color

(b) Interpolating the lighting intensity

Figure 21: Generalization of FC-StyleGAN by varying the downscaled resolution $\phi$ and interpolating one of the fine-grained factors. In the test image, we shift the robot position to the right-hand side, which is also attached with an unseen object (i.e., octahedron). We can see that in FC-StyleGAN with different downscaled resolutions, the considered factor keeps changing during its interpolations. Furthermore, the interpolated images in each case maintain the new robot position and particularly maintain new object shape (i.e., octahedron) in the case of $\phi = 64$. The reason why the new object shape is only maintained at $\phi = 64$ is beacuse that the object shape, together with the robot position, is not fine-grained at $\phi = 64$ any more. Therefore, it demonstrates that the disentanglement learning of FC-StyleGAN can generalize well to unseen novel test images.

