# OpenReview forum: "Disentangled GANs for Controllable Generation of High-Resolution Images"
_ICLR.cc/2020/Conference — Reject_

### Official Review · AnonReviewer2 · 2019-10-07
**Official Blind Review #2**

**Rating:** 3

**Review:**

Summary:

The paper is reasonably clear (though see some of my detailed comments on writing below).
The idea seems well-motivated and somewhat new though not revolutionary,
and the new data sets are nice (though see comments below about how I'm not qualified to evaluate them),
but I don't understand why the proposed algorithm wasn't evaluated on existing data sets as well,
and I don't understand why it wasn't compared against other algorithms that purport to do the same thing.
Barring certain exceptions (again see detailed comments), it seems like those two things are things
we ask of essentially all machine learning paper submissions, and I don't see why this paper is different?
I am open to being persuaded, but as of now I can't recommend acceptance.

Detailed Comments:

> More importantly, only using 5% of the labelled data significantly improves the disentanglement quality.
Hard to parse.
Does this mean using only 5% of data is better for disentangling than using 100%?


>  Generative adversarial networks (GANs) (Goodfellow et al., 2014) have achieved great success at generating realistic images, such as StyleGAN
Strange sentence - styleGAN is not itself a realistic image.


>  the controllable generation of high-resolution images is possible
Nit: i would cut the 'the'


> which characterizes how significant that the model can change a factor.
What does this mean?


> present a state-ofthe-art challenge
What does it really mean for a challenge to be state of the art?

> that enables conditional generation of high-fidelity images
Pretty minor nit, but I don't think it makes sense to refer to samples themselves as high fidelity.
High fidelity to what?
It makes a little more sense to refer to a trained generator as having high fidelity to the training data set,
but TBH I still don't even like the phrase in that context.

Fig 1 is good.
I felt like I could understand what's going on mostly from looking at the figure, which is nice.

The section surrounding eqs 1 and 2 is a little hard to read for me.
Lots of single letter variable names and it's hard to keep them all in my head when I read the equation.
I'd replace e.g. c_r with \text{code}_r and so forth.

> Note that AC-StyleGAN reduces to an InfoGAN variant in the special case
This is helpful.

> AC-StyelGAN

>  FCStyleAGN

> by symmetry
I don't follow this part.


> while its high-resolution blocks accounts for fine styles
I feel like it's worth making a distinction between high frequency bits of an image and non-essential (or nuisance variables or whatever) bits.
They often are the same, but not always (it really just depends how close up your photo is, right?) but this technique is really separating high-freq from low-freq, IIUC.

>  handling complex high-fidelity images
Again, I just don't feel like this phrase makes any sense.

Re eq 3:
I guess you can call this the 'interpolation variance'
if you want, but it's not really a new thing.
You're just getting discretized measurements of the Jacobian of the mapping from code to predicted code, and I don't really think in a way that fits with the
intuition you describe.
There are a few things I can think of that are weird about this measurement, but here's just one:
I might have a dimension of the code that causes one really peaky change at one point, and then the change quickly reverts.
If i measure the variance of the output, I won't see much (in fact, depending on how you discretize, maybe I'll miss it altogether),
even though the fact that the change is peaky doesn't say anything about the 'importance' of the change.
I think you want something more like a line integral.
It's possible I'm misunderstanding something about the description of this, however.

Re: the experiments:
I like the new data sets, although I'm not familiar enough w/ the robotics literature to know whether they add something really new.
What I don't understand is why there's not really any attempt to do either of:
a) compare the models from the paper to existing models
b) evaluate the models from the paper on existing data-sets?

I understand that sometimes one comes up with a thing that does something totally new,
and then reviewers will (maybe unjustifiably) still insist that the some kind of comparison be made even though it doesn't make sense,
but that doesn't seem to be what has happened here?
This paper proposes a new algorithm for synthesizing images and controlling various attributes of the images, but this has certainly been done in the past,
so why can't your technique be compared to those prior techniques?

Similarly, there are lots of data sets that people are already familiar with that you could have evaluated these models on, right?


**Experience Assessment:**

I have published in this field for several years.

**Review Assessment: Checking Correctness Of Derivations And Theory:**

N/A

**Review Assessment: Checking Correctness Of Experiments:**

I assessed the sensibility of the experiments.

**Review Assessment: Thoroughness In Paper Reading:**

I read the paper at least twice and used my best judgement in assessing the paper.

---

> ### Author Response · Authors · 2019-11-14
> **Response to Reviewer 2**
>
> Thank you very much for your review and helpful comments. We address your specific questions and comments below:
>
> 1. I don't understand why the proposed algorithm wasn't evaluated on existing datasets.
>
> - We have evaluated our model on dSprites, one of the most commonly disentanglement datasets. The results of comparing with the state-of-the-art unsupervised disentangled VAEs, such as Beta-VAE, FactorVAE and Beta-TCVAE are shown in Table 2(a), and we can see the unsupervised version of our model outperforms all the *generic* disentanglement learning baselines.
> - As the reviewer 3 suggested, we have also provided the latent traversal results of our model on CelebA of resolution 256x256 to demonstrate the effectiveness of our model on real datasets.
>
> 2. I don't understand why it wasn't compared against other algorithms that purport to do the same thing.
>
> - Thanks for the suggestion! We agree that there are many works on synthesizing images and controlling various attributes. But many of them are either domain-specific (only for faces [1] or person images [2] or 3D features [3]) or are not able to control the generation of *high-resolution* images (at least 512x512). To the best of our knowledge, there is little prior work on *semi-supervised* *generic* disentanglement learning of *high-resolution* images for us to systematically compare with.
> - More similarly, the disentangled VAEs, such Beta-VAE, FactorVAE and Beta-TCVAE are a class of unsupervised *generic* disentanglement learning models. By setting \alpha=0, the unsupervised version of our model can be compared with these disentangled VAEs. We have run experiments on a downsampled version of the Isaac3D dataset (with resolution 128x128), and found that the unsupervised version of our model outperforms all of these disentangled VAEs in terms of disentanglement quality and image quality. We have added these comparison results into Section 4.2 in the revised paper.
>
> [1] Tran et al., "Disentangled representation learning gan for pose-invariant face recognition." CVPR 2017.
> [2] Ma et al., "Disentangled person image generation." CVPR. 2018.
> [3] Nguyen-Phuoc et al., "HoloGAN: Unsupervised learning of 3D representations from natural images." ICCV 2019.
>
> 3. There are a few things I can think of that are weird about the measurement — 'interpolation variance'. For example, I might have a dimension of the code that causes one really peaky change at one point, and then the change quickly reverts.
>
> - We agree that generally if the number of interpolating points (denoted by S in Eq. (3)) is not large enough, we may miss some peaky changes. However, we didn’t observe this kind of peaky change in experiments, because the latent traversal results of our model are quite smooth. For example, we have increased the value of S in experiments, but the results of identifying fine-grained styles basically remain unchanged.
>
> 4. Other representations issues
>
> - We thank the reviewer for pointing them out. We have fixed most representation issues accordingly.
>
> Please let us know if we have addressed your concerns and if you have further comments.

---

### Official Review · AnonReviewer1 · 2019-10-23
**Official Blind Review #1**

**Rating:** 3

**Review:**

=== A. Summary ===

This paper proposes to train a new conditional GAN model that allows for controllable image generation by changing the input factors of variations (e.g. object color).
The supervised labels for the controllable attributes are obtained from a 3D renderer.
That is, the work combines the recent StyleGAN (that learns to generate images with disentangle latent vectors in an unsupervised manner) with AC-GAN (a clas-conditional GAN but here class information is replaced by the attribute information that we want to control).
The resultant AC-StyleGAN has essentially two latent vectors, one trained unsupervised and one trained with supervised labels.

The proposed GANs were thoroughly tested with different factors of variations (lighting, camera, objects) and on two different datasets self-contructed via 3D renderer.
The work is a solid demonstration that GANs can be used to synthesize images with fine-grained and coarse controllability if we have supervision signals!
The authors also released the anonymous code (which is a plus!).


=== B. Decision ===

Weak Reject.

I voted for Weak Reject because this paper presents a fairly incremental advance over what has been done (e.g. HoloGAN, StyleGAN, AC-GAN).
The impact of the unsupervised work (e.g. HoloGAN or StyleGAN) is much higher since they are generally applicable to real data without labels.
Here, although reasonable, the demonstration was done on a relatively small-scale 3D synthetic data (where there is only one scene and one object being manipulated).
Therefore, the claim that only 5% of supervised labels is required may not carry over to larger-scaled datasets with larger scene variability.
Plus, I don't see any baseline whatsoever being compared with the proposed methods here.


=== C. Suggestions for Improvement ===

I have no problems with the novelty or idea of the work.
For me, the key problem with this work is the low impact or significance.

Some suggestions for showing the impact of this work:
- Show how your pre-trained GANs can be fine-tuned or transferred to the real data where we don't have labels.
- You could also plug in your pre-trained GANs to a separate synthetic-to-real image translation model to show that we could indeed learn to control these factors of variations of the real images. Hopefully, would the above setups yield better results than HoloGAN or StyleGAN?
- Clarify the main focus points of this paper and try to substantiate the result. The author wrote "Our work extends the above works by scaling up the disentanglement learning to high- resolution images, and emphasizing the importance of supervision in controllable generation." <---- but (1) high-res images here are synthetic and limited in scene variability; (2) the second part is expected given previous work.

**Experience Assessment:**

I have published in this field for several years.

**Review Assessment: Checking Correctness Of Derivations And Theory:**

I assessed the sensibility of the derivations and theory.

**Review Assessment: Checking Correctness Of Experiments:**

I assessed the sensibility of the experiments.

**Review Assessment: Thoroughness In Paper Reading:**

I read the paper at least twice and used my best judgement in assessing the paper.

---

> ### Author Response · Authors · 2019-11-14
> **Response to Reviewer 1**
>
> Thank you very much for your review and helpful comments. We address your specific questions and comments below:
>
> 1. This paper presents a fairly incremental advance over what has been done (e.g. HoloGAN, StyleGAN, AC-GAN). The impact of the unsupervised work (e.g. HoloGAN or StyleGAN) is much higher since they are generally applicable to real data without labels.
>
> - We agree that the *unsupervised* disentanglement learning on real data, such as HoloGAN, is a very interesting direction. Mostly, the model inductive bias introduced in these unsupervised methods makes them 1) only applicable to a specific domain, and/or 2) difficult to scale up to high-resolution images.
> - On the contrary, the semi-supervised method can apply to a *generic* disentanglement learning, and it could also be of high practical impact if we only need to use a very limited portion of labelled data. In such sense, our work is complementary to these previous unsupervised methods.
>
> 2. The claim that only 5% of supervised labels is required may not carry over to larger-scaled datasets with larger scene variability.
>
> - This is a good point! We share the same concern that with more scene variability, more supervision may be required to achieve the same disentanglement quality. This motivates the creation of two new disentanglement datasets, which are of higher variability and resolution than the existing disentanglement datasets as listed in Table 1 of the revised paper. We think our datasets and results can serve as a good starting point in the direction of scaling up the *generic* disentangling learning to large-scale real data. Based on our work, there is more future work to be done to keep increasing the scale and variability of the disentanglement datasets and models.
>
> 3. I don't see any baseline whatsoever being compared with the proposed methods here. Some suggestions: 1) Show how your pre-trained GANs can be fine-tuned or transferred to the real data where we don't have labels. 2) You could also plug in your pre-trained GANs to a separate synthetic-to-real image translation model to show that we could indeed learn to control these factors of variations of the real images. 3) Compare with HoloGAN and StyleGAN.
>
> - These are great suggestions, but the use of transfer learning or a separate synthetic-to-real image translation model is out of the scope of this paper. This is because doing so will introduce another research topic about the cross-domain disentanglement learning, but our main focus is on the *generic* disentanglement learning with deep generative models.
> - Regarding comparison with HoloGAN, we do not think it is a fair comparison: HoloGAN is restricted to only disentangling 3D representations for relatively low-resolution images, however, our model is a generic semi-supervised disentanglement learning model especially designed for controlling high-resolution images.
> - Regarding comparison with StyleGAN, we have already use InfoGAN modified StyleGAN (when \alpha=0 in Eq (2)) as a stronger unsupervised baseline than the original StyleGAN.
> - Regarding the evaluation on real images, we have run our model on CelebA of resolution 256x256 as Reviewer 3 suggested and showed the good latent traversal results in Section 4.2 of the revised paper.

---

> > ### Author Response · Authors · 2019-11-14
> > **Response to Reviewer 1 (cont.)**
> >
> > 4. Clarify the main focus points of this paper and try to substantiate the result. The author wrote "Our work extends the above works by scaling up the disentanglement learning to high-resolution images, and emphasizing the importance of supervision in controllable generation." <---- but (1) high-res images here are synthetic and limited in scene variability; (2) the second part is expected given previous work.
> >
> > - In the *generic* disentanglement learning community, most commonly used benchmarks are synthetic datasets, such as dSprites and Shape3D (see more in [1]). This is because the factors of variation are *known* in the synthetic datasets and thus we can quantify the disentanglement quality of different models. We think it is good to first study more complex synthetic datasets to gain insights and then apply the insights to real datasets. Furthermore, we have tested a higher-resolution version of CelebA (compared with most previous methods working on 64x64 CelebA) and confirmed the effectiveness of our model on high-resolution real datasets.
> > - As far as we know, no much work on the *generic* disentanglement learning has systematically investigated the importance of *limited* supervision, except for [2]. However, [2] only applies to disentangled VAEs on low-resolution synthetic datasets, raising another open question: With very little supervision on more complex and higher-resolution datasets, can we still well disentangle the factors of variation while maintaining the high generation quality? One major goal in our work is to answer this question with a thorough experimental study. We have highlighted this point in the revised paper.
> > - Finally, we have slightly modified the abstract and introduction for clarifying the main focus points of this paper as the reviewer suggested.
> >
> > [1] Locatello et al., “Challenging Common Assumptions in the Unsupervised Learning of Disentangled Representations”. ICML 2019.
> > [2] Locatello et al., “Disentangling factors of variation using few labels”. arXiv preprint arXiv:1905.01258.
> >
> > Please let us know if we have addressed your concerns and if you have further comments.

---

### Official Review · AnonReviewer3 · 2019-10-28
**Official Blind Review #3**

**Rating:** 3

**Review:**

Disentangled GANs for controllable generation of High-Resolution Images introduces two new high resolution synthetic scene datasets for studying disentanglement in generative models and benchmarks two Style-GAN based architectures for controllable generation on these datasets. The datasets, though still synthetic, provide a significant quality boost over some of the simpler toy datasets previously studied in the disentanglement literature. A variety of experiments are conducted looking at 3 metrics (FID, MIG, and latent reconstruction) as well as qualitative analysis of samples. The paper studies how the performance of these architectures varies with hyperparameter and design decisions. The authors demonstrate that AC-StyleGAN performs well in fully supervised settings achieving the desired conditional generation, but, when controlling only a subset of factors, does not correctly disentangle. To address this, the author’s other architecture modification, FC-StyleGAN an image to image model is demonstrated to improve performance in this setting. The paper provides the reader with a useful overview of the behavior of the two models on these two datasets.

My rating is weak reject. While the paper has a variety of contributions (most notably introducing two new datasets and modifications of StyleGAN) and interesting results (such as relatively high performance with only 5% labeled data and the disentanglement issues when controlling a subset of factors), the core contributions of new datasets and architectures are not validated or analyzed rigorously enough.

1) No prior work is used as baselines in order to compare / validate the newly introduced architectures. The authors dismiss prior work in the introduction but do not provide any direct evidence that prior work is unable to handle the datasets introduced in the paper. In order to consider acceptance based on the value of the architectures proposed in the paper, there should be some direct evidence that the proposed Style-GAN modifications AC/FC, outperform prior architectures. One particular choice which is unclear to the reviewer is why AC GAN was chosen over the Projection Discriminator of Miyato and Koyama 2018 which demonstrated significantly better results than AC GAN and was adopted by major followup work such as BigGAN.

2) Given the core contributions of new architectures for disentanglement, the lack of any results on real, non synthetic, datasets in order to validate these (for instance, CelebA) which significant prior work on disentanglement has been evaluated on is an unfortunate omission. It is unclear how the introduced architectures handle complex natural image distributions with the much more diverse sources of natural variation they contain.

3) The authors claim their method is effective in the semi-supervised setting and demonstrate this by showing relatively strong performance in a low data regime. This demonstrates the importance of the third term in the loss function, but as other work for semi-supervised learning has shown (Oliver et al 2018) care must be take to properly attribute the performance of a semi-supervised algorithm to the actual semi-supervised components and purely supervised baselines are often quite competitive. The addition of an ablation / analysis demonstrating the contribution of the second term in settings where labels are also available would strengthen the author's claims and fully demonstrate the model is an effective semi-supervised learner.

Additional comments:

The datasets introduced in the paper do seem like potentially valuable contributions to the community - more discussion on the motivations behind their creation, the differences with prior work, the open difficulties / challenges, as well as the recommended evaluation protocols could add to their value.

The authors could more clearly motivate the work / applications the paper is interested in and how each contribution / experiment fits in with this. Without a clear sense of the authors mission / goals with the work, it feels a bit difficult to interpret the results.



**Experience Assessment:**

I have published in this field for several years.

**Review Assessment: Checking Correctness Of Derivations And Theory:**

N/A

**Review Assessment: Checking Correctness Of Experiments:**

I assessed the sensibility of the experiments.

**Review Assessment: Thoroughness In Paper Reading:**

I read the paper at least twice and used my best judgement in assessing the paper.

---

> ### Author Response · Authors · 2019-11-15
> **Response to Reviewer 3**
>
> Thank you very much for your review and helpful comments. We address your specific questions and comments below:
>
> 1. The core contributions of new datasets and architectures are not validated or analyzed rigorously enough.
>
> - We have included Table 1 to directly summarize the advantages of our proposed datasets over previous datasets.
> - We have also slightly modified the abstract and introduction for clarifying the main focus points of this paper as the reviewer suggested.
> - To further validate our contributions, we have also done the ablation studies on semi-supervised learning, compared with baselines on both the existing dataset and our proposed dataset, and applied our method to real data.
>
> 2. No prior work is used as baselines in order to compare / validate the newly introduced architectures. Especially, why AC-GAN was chosen over the Projection Discriminator of Miyato and Koyama 2018.
>
> - We choose AC-GAN, instead of other conditional GAN methods, such as cGANs (with Projection Discriminator), mainly because only AC-GAN has the property of reconstructing meta code in the discriminator, which can easily be extended to a semi-supervised disentanglement learning framework.
> - We indeed did experiments on comparing AC-GAN, vanilla cGAN and cGANs with Projection Discriminator in the fully supervised case. Our results show that the image quality generated by AC-GAN modified StyleGAN largely outperforms the other two methods. We have made it more clear in the revised paper that only AC-GAN framework can support *semi-supervised* disentanglement learning.
>
> 3.  The lack of any results on real, non synthetic, datasets in order to validate these (for instance, CelebA) which significant prior work on disentanglement has been evaluated on is an unfortunate omission.
>
> - One major motivation is that how (semi-)supervision helps disentanglement. In such sense, we need the ground-truth meta code to provide (semi-)supervision and *quantify* our results. However, most real datasets lack the ground-truth meta code for factors of variation. It is why we want to focus on the synthetic datasets with quantitative results to gain useful insights.
> - We agree that many previous works use CelebA to evaluate the disentanglement quality. But they can only show the *qualitative* results of latent traversal since there are only *binary* attributes in CelebA. Commonly-used disentanglement metrics, such as MIG and FactorVAE score, do not apply to factors of *binary* variations.
> - As the reviewer suggested, we have added latent traversal results on CelebA in the revised paper to qualitatively validate our model in real datasets, as shown in Figure 5 and Figure 15.
>
> 4. The addition of an ablation / analysis demonstrating the contribution of the second term in settings where labels are also available would strengthen the author's claims and fully demonstrate the model is an effective semi-supervised learner.
>
> - Thanks for the suggestion. We have run the suggested experiments by removing the unsupervised disentanglement term in the discriminator loss function. This serves as a supervised baseline by only using the available labelled data. We found that when only 1% or 5% of labelled data is available, our semi-supervised method consistently outperforms the supervised baseline, as shown in Table 3.
>
> Please let us know if we have addressed your concerns and if you have further comments.

---

### Author Response · Authors · 2019-11-15
**Major changes in the revised paper**

We thank all the reviewers for very useful comments and suggestions. Below are majors changes in the revised paper according reviewers’ suggestions:

- We want to emphasize that the motivation of this work is to investigate 1) how a *generic* disentanglement learning model behaves in the *high-resolution* image domain and 2) how the limited supervision could help the disentanglement. We have highlighted these two points in the revised paper.

- We have added a rescaling coefficient in the semi-supervised loss function Eq. (2), and obtained *better* semi-supervised results in Figure 3: By only use *1%* labelled data, the disentanglement quality is close to the fully supervised case and outperforms the unsupervised alternative by a significant margin.

- We have added a comparison between our model and disentangled VAEs on both the dSprties and the downsampled version of the Isaac3D dataset in Table 2. We found that the unsupervised version of our model largely outperforms these baselines in terms of both image quality and disentanglement quality.

- We have added the experimental results on CelebA with resolution 256x256 and showed the effectiveness of our model on the real data  in Figure 5 and 15.

- We have added a supervised baseline which only uses the labelled data in the disentanglement learning. The results in Table 3 showed our semi-supervised method consistently outperforms the supervised baseline in the case where only 1% or 5% labelled data is available.

---

### Decision · Program_Chairs · 2019-12-19

**Decision:**

Reject

**Comment:**

The paper presents a model combining AC-GAN and StyleGAN for semi-supervised learning of disentangled generative adversarial networks. It also proposes new datasets of 3d images as benchmarks. The main claim is that the proposed model can achieve strong disentanglement property by using 1-5% of the annotations on the factors of variation. The technical contribution is moderate but the architecture itself is not highly novel. While the proposed method seems to work for controlled/synthetic datasets, overall technical contribution seems incremental and it's unclear whether it can perform well on larger-scale, real datasets. The experimental results on CelebA don't look convincing enough.